# Genomic and single-cell analyses reveal genetic signatures of swimming pattern and diapause strategy in jellyfish

Zhijun Dong [1,2,8] ✉, Fanghan Wang [1,2,8], Yali Liu[2,3,4,8], Yongxue Li[1,2,8], Haiyan Yu[5], Saijun Peng[1,2], Tingting Sun[1,2], Meng Qu[2,3,4], Ke Sun[6], Lei Wang[1,2], Yuanqing Ma[7], Kai Chen [6], Jianmin Zhao [1,2] ✉ & Qiang Lin [2,3,4] ✉

Jellyfish exhibit innovative swimming patterns that contribute to exploring the origins of animal locomotion. However, the genetic and cellular basis of these patterns remains unclear. Herein, we generated chromosome-level genome assemblies of two jellyfish species, *Turritopsis rubra* and *Aurelia coerulea*, which exhibit straight and free-swimming patterns, respectively. We observe positive selection of numerous genes involved in statolith formation, hair cell ciliogenesis, ciliary motility, and motor neuron function. The lineage-specific absence of otolith morphogenesis- and ciliary movement-related genes in *T. rubra* may be associated with homeostatic structural statocyst loss and straight swimming pattern. Notably, single-cell transcriptomic analyses covering key developmental stages reveal the enrichment of diapause-related genes in the cyst during reverse development, suggesting that the sustained diapause state favours the development of new polyps under favourable conditions. This study highlights the complex relationship between genetics, locomotion patterns and survival strategies in jellyfish, thereby providing valuable insights into the evolutionary lineages of movement and adaptation in the animal kingdom.

Jellyfish represent one of the most critical evolutionary lineages of the animal kingdom and play a central role in the evolution of early animal movement systems[1,2]. Locomotion drives several adaptive biological traits, such as balance, pressure perception, and orientation, all of which contribute to the achievement of neuromuscular control of movement[3]. Maintaining balance is crucial for ensuring stability during movement and typically involves coordination between sensory and motor systems[4]. In Eumetazoans, the structures responsible for balance range from simple statocysts in aquatic invertebrates to complex inner ears in mammals[5]. These structures consist of mass blocks of calcium crystals, proteoglycans, and collagen. Additionally, they also comprise sensory hair cells that are mechanically influenced by the position of the mass blocks. It is possible that these structures may have been present in the last common ancestor of bilaterians, cnidarians, and ctenophores[6,7] (Supplementary Fig. 1).

[1]CAS Key Laboratory of Coastal Environmental Processes and Ecological Remediation, Yantai Institute of Coastal Zone Research, Chinese Academy of Sciences, Yantai, Shandong 264003, China. [2]University of Chinese Academy of Sciences, Beijing 100101, China. [3]CAS Key Laboratory of Tropical Marine Bio-Resources and Ecology, South China Sea Institute of Oceanology, Chinese Academy of Sciences, Guangzhou 510301, China. [4]Guangdong Provincial Key Laboratory of Applied Marine Biology, South China Sea Institute of Oceanology, Chinese Academy of Sciences, Guangzhou 510301, China. [5]College of the Environment and Ecology, Xiamen University, Xiamen 361102, China. [6]State Key Laboratory of Primate Biomedical Research, Institute of Primate Translational Medicine, Kunming University of Science and Technology, Kunming, Yunnan 650500, China. [7]Shandong Key Laboratory of Marine Ecological Restoration, Shandong Marine Resource and Environment Research Institute, Yantai, Shandong 264006, China. [8]These authors contributed equally: Zhijun Dong, Fanghan Wang, Yali Liu, Yongxue Li. ✉e-mail: zjdong@yic.ac.cn; jmzhao@yic.ac.cn; linqiang@scsio.ac.cn

In jellyfish, a statocyst is located on top of each sensory hair cell and functions as a feedback system by responding to gravity and regulating orientation[8]. Statocyst removal results in a loss of orientation and inability to perform righting movements[9]. As a rare exception, statocysts are absent in Anthomedusae, which also contain medusa stages[10]. The swimming patterns of the ellipsoidal jellyfish, *Turritopsis rubra*, and the oblate jellyfish, *Aurelia coerulea*, are characterised by a typical jet propulsion (straight swimming) and rowing propulsion mechanisms (free swimming), respectively (Fig. 1a)[11]. In Anthomedusae, a network of pacemakers to control swimming contractions is located in the marginal nerve rings, with light-sensitive structures (ocelli) distributed around the margin. In Scyphomedusae and Cubomedusae, swimming pacemakers are restricted to marginal integration centre called rhopalia, which contain ocelli and statocysts (Fig. 1b), and damage or experimental removal of the rhopalia reduces the overall speed and regularity of swimming[12]. Although statocysts, hair cells, and their relatives determine swimming patterns in jellyfish, the genetic basis underlying the occurrence and formation of jellyfish movement signatures remains unclear. Statocysts, in particular, provide interesting insights into the specific locomotion patterns of jellyfish. Among

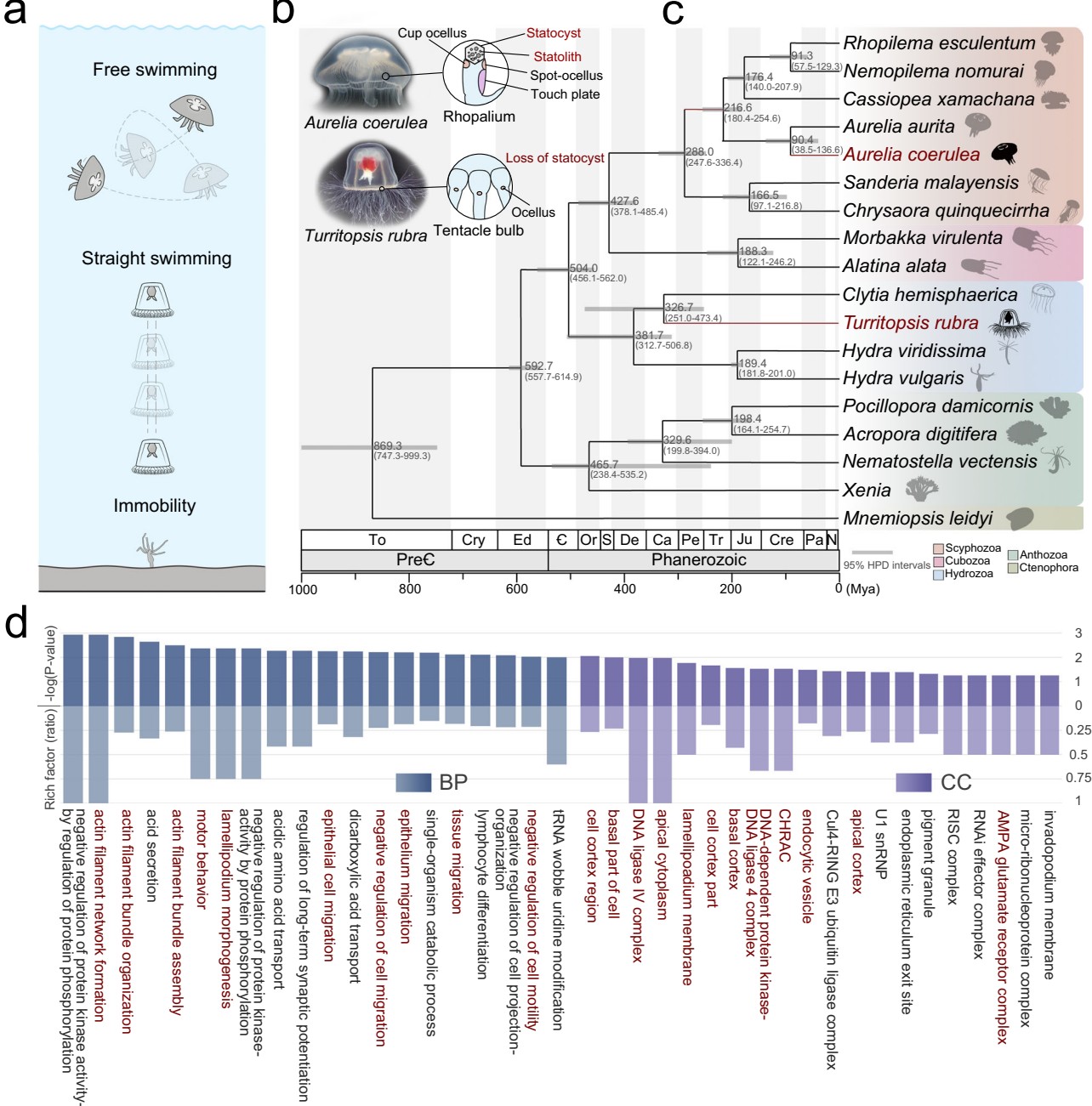

**Fig. 1 | Evolution and swimming features of *Turritopsis rubra* and *Aurelia coerulea*. a** Movement modes of *Hydra vulgaris*, *T. rubra*, and *A. coerulea*, ranging from immobility to free swimming. **b** Images and schematics of the sensory organs of *A. coerulea* (rhopalium) and *T. rubra* (tentacle bulb). **c** Phylogenetic tree of 17 cnidarians and a ctenophore as the outgroup. The bootstrap value of all nodes is 100. **d** The top 20 enriched Gene Ontology (GO) terms for positively selected genes (PSGs) in the *T. rubra* genome are shown for biological processes (BPs) and cellular components (CCs). Categories involved in statocyst formation and movement are coloured in red. The enrichment was conducted using the GOseq R package, and corrected *P* < 0.05 indicated significant enrichment. Source data are provided as a Source Data file.

the early branching species, jellyfish were an early and simple species to employ striated muscle and a nervous system to regulate autonomous swimming[13], rendering them as unique metazoans for investigating the mechanisms underlying locomotor evolution.

In nature, the evolution of life forms has led to the emergence of diverse survival strategies[14]. In jellyfish, the evolution of locomotion has resulted in a range of adaptations and swimming strategies, such as ambush and cruising foraging patterns. Passive drifting is a characteristic of ambush predators and usually manifests as straight swimming, whereas mobility is crucial for cruising predators and usually manifests as free swimming[11]. Changes in motility allow jellyfish to adapt to environmental conditions and play a key role in the survival of the species, especially when food is scarce[15]. The straight-swimming species that struggle to quickly escape under adverse conditions may evolve survival strategies to reduce energy expenditure and wait for better conditions, which would become crucial for their survival.

Jellyfish exhibit unidirectional alternation between the stationary phase (asexual reproduction) and motile phase (sexual reproduction)[16]. However, some species such as *T. dohrnii*, can reverse their development from sexually reproducing jellyfish to asexual polyps when subjected to prolonged starvation, and are therefore, known as 'immortal jellyfish'[17]. However, the existence of reverse development in other species, such as *T. rubra*, remains ambiguous as their complete life cycle has not been conclusively discerned within natural habitats[18]. Although some work has been conducted to explore the mechanisms of reverse development at the genomic and transcriptomic levels[19,20], the evolutionary signatures underlying the swimming pattern and survival strategy in jellyfish remain unclear. Hence, the genetic and cellular basis of jellyfish development, particularly reverse developmental stages, could enhance our understanding of the distinctive life strategies of jellyfish.

In this study, we aimed to elucidate the functional alterations in statocysts and hair cells during the evolutionary trajectory of jellyfish locomotion, and sequenced high-quality, chromosome-level genomes of two jellyfish species, *T. rubra* and *A. coerulea*. These genomes were subsequently juxtaposed with previously documented cnidarian genomes, focusing on the evolutionary innovations and genetic signatures associated with statocysts and hair cells in species characterised by divergent swimming modes. The disparities in gene transcription dynamics and cellular transitions across the jellyfish species were investigated. A comprehensive single-cell-resolution transcriptomic analyses of the five stages of normal and reverse development in *T. rubra* was conducted to identify the cell types and gene expression patterns across diverse developmental stages.

## Results

### Jellyfish genomic features and phylogenomic analyses

We generated two high-quality, chromosome-level reference genomes of the hydrozoan jellyfish, *T. rubra* (2n = 30), and the scyphozoan jellyfish, *A. coerulea* (2n = 44), spanning 267 Mb and 566 Mb and containing 18,746 and 32,035 gene annotations, respectively. Pseudo-chromosome syntenic analyses demonstrated that the genome structure of *T. rubra* was highly conserved with that of *A. coerulea* (Supplementary Fig. 5). These are the chromosome-level genome assemblies for *Turritopsis* and *Aurelia* and have higher complete Benchmarking Universal Single-Copy Orthologue (BUSCO) values and genome coverages compared to previously published genomes[19–22], suggesting higher assembly qualities (Supplementary Table 13).

To determine the phylogenetic position of *T. rubra* and understand the emergence of balancing organs and the adaptive evolution of locomotor patterns, we performed a phylogenetic analyses using whole-genome datasets of 17 cnidarians (from immobilised anthozoans to free-swimming scyphozoans), with one ctenophore included as an outgroup (Fig. 1c). In accordance with previously published phylogenetic reconstructions[21], the separation of the major cnidarian clades,

estimated using molecular dating, occurred approximately 600 million years ago (Mya). Our phylogenomic analyses indicated that *T. rubra* was a sister taxon of *Clytia hemisphaerica* within the Hydrozoa clade, with maximal bootstrap support. The species divergence times estimated using calibration points revealed that *T. rubra* and *C. hemisphaerica* diverged approximately 326.7 Mya [95% highest posterior density (HPD) 251.0–473.4 Mya].

### Genetic basis of statocyst and movement regulation

To investigate the genetic basis of the adaptive evolution of locomotion, we performed a comparative genomic analyses of *T. rubra* and seven other jellyfish species (*C. hemisphaerica*, *Morbakka virulenta*, *Sanderia malayensis*, *A. coerulea*, *Cassiopea xamachana*, *Nemopilema nomurai*, and *Rhopilema esculentum*). A total of 548 positively selected genes (PSGs) enriched for epithelial cell migration, motor behaviour, and tissue migration were identified (Supplementary Data 1 and 2). Of these, PSGs related to cilium assembly and movement (actin filament bundle, motor behaviour) that are essential for statocyst formation[23], may contribute to the absence of a statocyst in *T. rubra* (Fig. 1d, Supplementary Data 3). Alternatively, genes enriched in the AMPA glutamate receptor complex may be involved in the nerve-mediated regulation of movement[24].

Statolith, a small, stone-like object that rests on the bristles of sensory cells (hair cells), is crucial for jellyfish movement and orientation[8]. Scanning electron microscopy of the tentacle bulbs of *T. rubra* and the rhopalia of *A. coerulea* revealed that in the former, sensory areas were characterised by hair cells carrying only one long kinocilium on their surface. In contrast, in the statocysts of *A. coerulea*, hair cells carried long, straight motile kinocilia surrounded by short crowns (a sort of folded crater) of non-motile stereocilia around the central shaft (Fig. 2a). Genes involved in regulating statolith formation, cilium morphogenesis and movement, including *CHSY1*, *GNPTAB*, *LOXHD1*, and *USH2A*, were also positively selected in *T. rubra* (Fig. 2b).

Swim motor neurons are located in the inner nerve ring, along each side of the radial canals and in the outer nerve ring, forming a centre that is mainly concerned with integrating sensory information and functioning with muscles to regulate the movement of *T. rubra*[25] (Fig. 2c). Some PSGs were specifically expressed in the nerve cells of *A. coerulea* but not *T. rubra*, such as *DCTN1*, *NRP1A*, and *KIF13B*, that are involved in regulating movement via nerves and muscles, indicating their essential roles in the evolution of the movement pattern observed in *A. coerulea* (Fig. 2c).

Transcriptome analyses of sensory organs and bell margins (controls) in four species (*Chrysaora quinquecirrha*, *R. esculentum*, *A. coerulea*, and *T. rubra*) also indicated significant differences in the expression of statocyst-related genes (Fig. 2d, Supplementary Data 4 and 5). Genes involved in normal otolith morphogenesis (*LRIG3* and *NOTUM2* implicated in otolith biomineralisation), ciliogenesis (*KIF14*), ciliary movement (*NPC1*, *FAM166B*, and *DNAH*s), and hair cell function (*LOXHD1* and *LRP5*) were highly expressed in the rhopalia of three scyphozoan jellyfish (Fig. 2d, e, Supplementary Fig. 8). The tentacle bulbs of *T. rubra* exhibited downregulation of genes involved in motile cilia (Fig. 2f, Supplementary Fig. 9, Supplementary Data 6). We also identified differentially expressed genes (DEGs) enriched in the sensory organs of each species (compared with the control samples) (Supplementary Fig. 10, Supplementary Table 16, Supplementary Data 7–9). These results suggest that genes related to nerves and sensory organs (eyes or ears) were up-regulated in respective sensory organs of each species, indicating a functional homology between jellyfish ocellus and statocysts and vertebrate eyes and ears. Furthermore, in *T. rubra*, various genes associated with motile cilia were found to be down-regulated. These genes are part of clusters related to cilia- and flagella-associated proteins (CFAPs), dynein axonemal heavy chains (DNAHs), intraflagellar transports (IFTs), and kinesins (KIFs). Interestingly, in the other three jellyfish

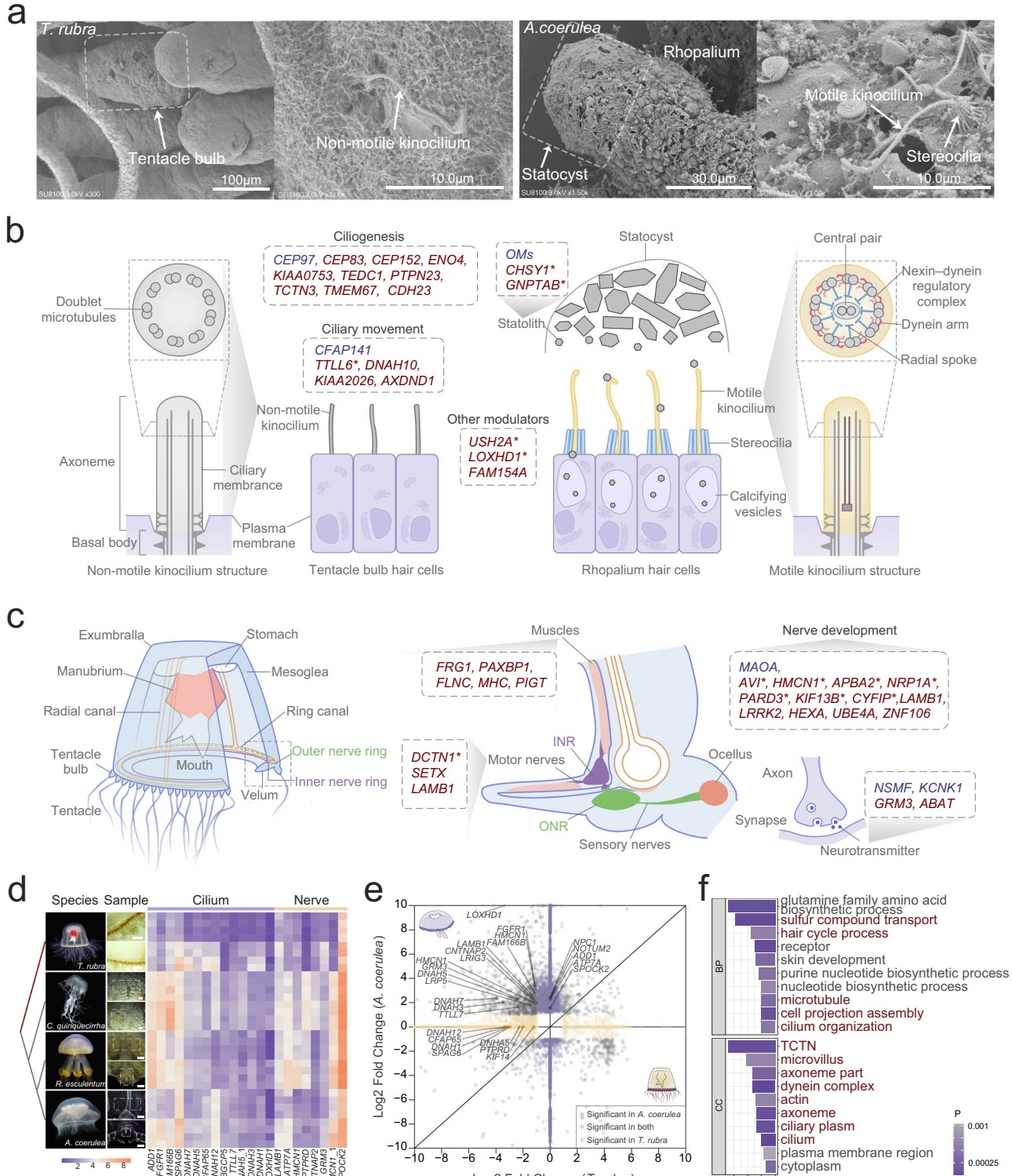

**Fig. 2 | Genetic features related to movement and lack of a statocyst in *Turritopsis rubra*. a** Scanning electron micrographs of sensory organs and cilium structures for *T. rubra* (left) and *Aurelia coerulea* (right). **b** Lost genes (blue) and positively selected genes (PSGs; red) related to statocyst formation and ciliary movement. Genes involved in otolith development reported in previous studies are marked with an asterisk. **c** Lost genes (blue) and PSGs (red) are involved in nerves and muscles that may regulate locomotion in *T. rubra*. Genes specifically expressed only in the neuronal cells of *A. coerulea* are marked with an asterisk. **d** Heatmap of differentially expressed orthologues related to cilia and nerves. Sequencing samples of each species are shown on the left for sensory organs (top) and control

tissue (bottom). Scale bar: 500 μm. **e** Volcano map of shared and group-specific differentially expressed genes (DEGs) in the sensory organ and control samples of *A. coerulea* compared with those of *T. rubra*. Purple, yellow, and grey represent markedly changed DEGs in *A. coerulea*, *T. rubra*, and both groups, respectively.
**f** Top 10 enriched Gene Ontology (GO) terms [biological processes (BPs) and cellular components (CCs)] for down-regulated DEGs in the tentacle bulb of *T. rubra* relative to those in the rhopalia of *A. coerulea*. Categories involved in cilium and statolith formation are coloured in red. The enrichment was conducted using the GOseq R package, and corrected $P < 0.05$ indicated significant enrichment. Source data are provided as a Source Data file.

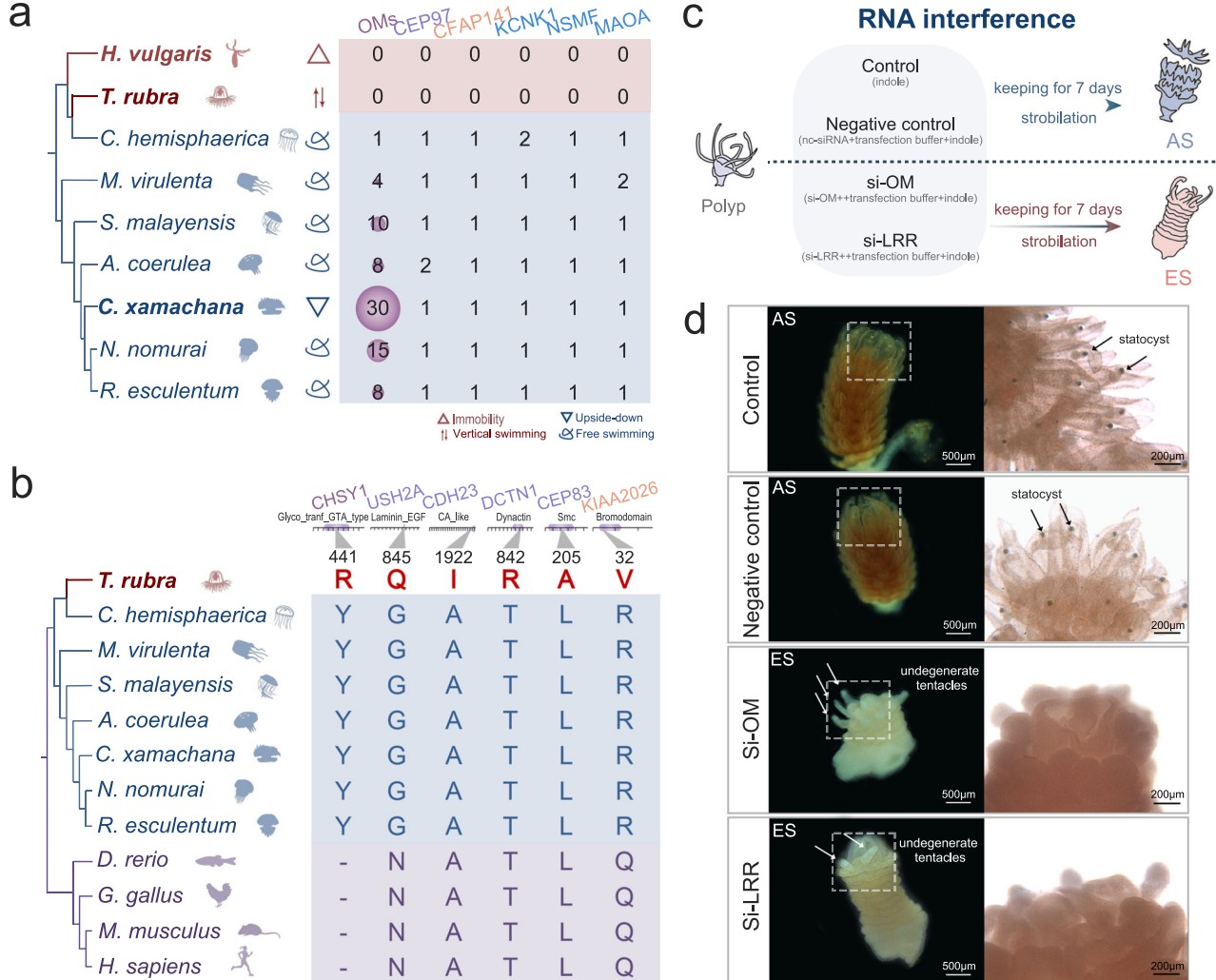

**Fig. 3 | Genomic comparison of features of lost genes and positively selected genes (PSGs) in *Turritopsis rubra*. a** Numbers of lost gene families related to statocysts (dark purple), ciliogenesis (light purple), ciliary movement (orange), and nerves (blue) in *T. rubra*. Swimming patterns of each species are shown. **b** Alignment of PSGs involved in the statocyst formation (statocyst in dark purple, ciliogenesis in light purple, and ciliary movement in orange) amino acid sequences of *T. rubra*, seven other cnidarians and four vertebrates, and genes exhibiting specific mutations (red) in *T. rubra*. The images of chordates were downloaded from BioRender.com. **c** Schematic diagram of the RNA interference experiment in *Aurelia coerulea*. ES early stage of strobilation, AS advanced stage of strobilation. **d** *A. coerulea* with knocked down OM genes (and OM proteins containing LRR structural domains) in ES with undegenerate tentacles, and control individuals in AS with long lappets and well-developed statocysts.

species, these same genes were up-regulated. These results indicate that the low expression of motile cilia-related genes may result in non- or dysfunctional cilia in sensory organs, contributing to the loss of statocysts in *T. rubra*.

**Absent and positively selected genes involved in the swimming pattern of *T. rubra***

A total of 278 gene families that underwent marked contractions and 156 gene families that were lost in the *T. rubra* genome were identified (Supplementary Fig. 6 and Supplementary Data 10–13). Gene family annotated as 'otolith morphogenesis' (OM) was observed in both the contracted and lost gene families (Fig. 3a). The OM family was predicted to comprise leucine-rich repeat-containing (LRR) proteins required for ciliary motility and otolith biogenesis[26,27] (Supplementary Fig. 12a). Furthermore, the loss of *CFAP141* and *CEP97* in *T. rubra* may also result in the absence of statoliths, as ciliary motility is required for normal otolith assembly and localisation[28] (Fig. 3a). The loss of *KCNK1*, *NSMF*, and *MAOA*, which are involved in the regulation of the locomotor network by the nervous system, may also contribute to the unique swimming patterns of *T. rubra*[29–31].

To explore the genetic evolution of statolith and otolith formation, the scope of the data was manually extended to a broader selection of taxa, including mammals, birds, and fishes. Alignment of the amino acid sequences of the PSGs mentioned earlier, which are involved in statocyst formation (*CHSY1*), ciliogenesis (*USH2A*, *CDH23*, *DCTN1*, and *CEP83*), and ciliary movement (*KIAA2026*), revealed that mutations in the conserved sites (Fig. 3b, Supplementary Fig. 12b) may contribute to altered gene functions and the inability to form statocyst in *T. rubra*.

We further conducted in situ hybridisation analyses to confirm whether these genes were involved in statocyst formation. *LOXHD1* was expressed exclusively in the area surrounding the statocyst in *A. coerulea* but not in *T. rubra* (Fig. 4c). Similarly, *USH2A* was expressed in the area surrounding the rhopalium in *A. coerulea* and the area surrounding the stomach in *T. rubra* (Fig. 4c). Two lost genes, OMs and *CFAP141* were specifically expressed in the area surrounding the statocyst in *A. coerulea* (Fig. 4c), indicating their role in statocyst formation.

To investigate the functional impact of OM genes on statocyst development, we knocked down the OM genes via short interfering

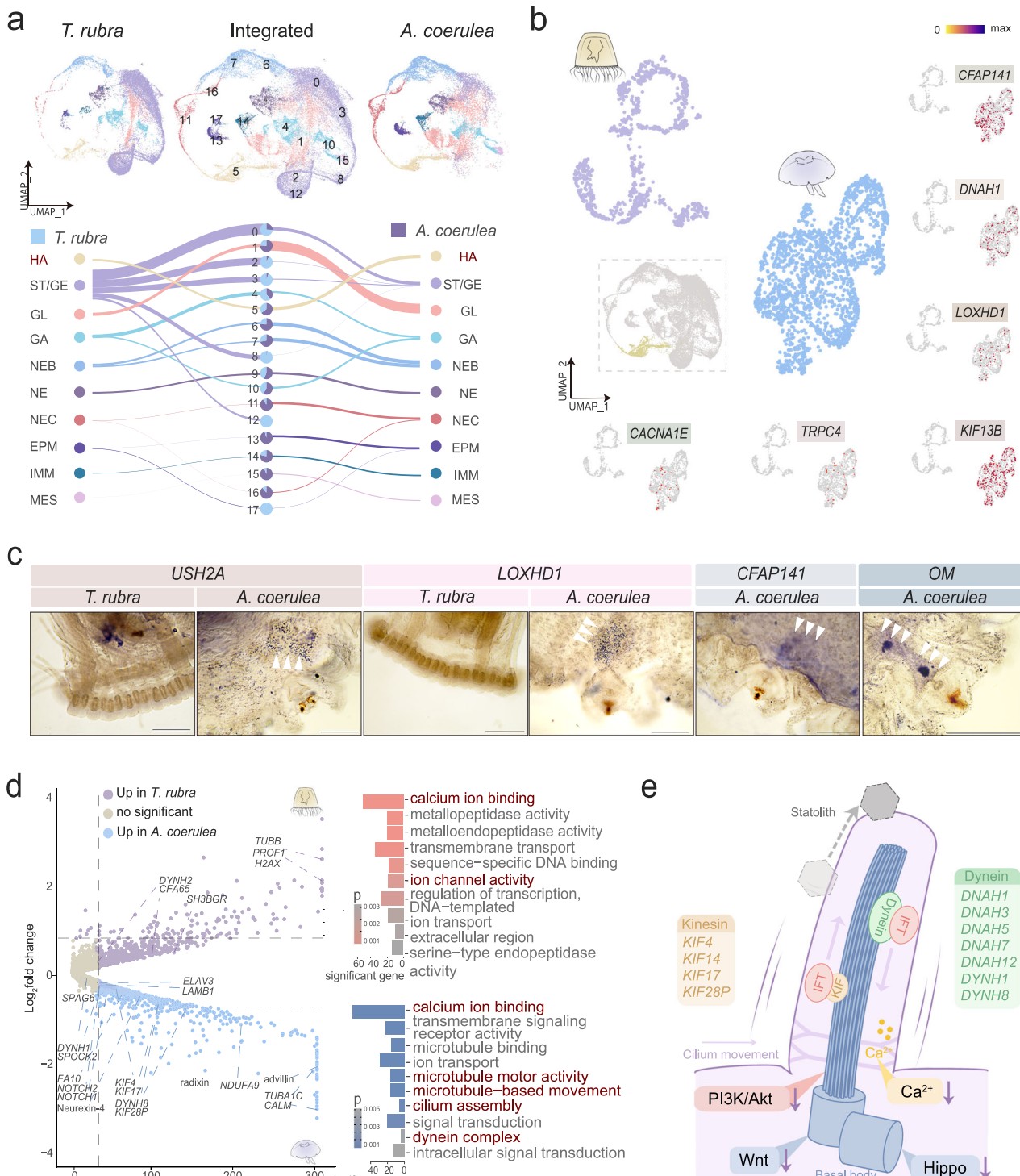

**Fig. 4 | Cross-species comparison of hair cells between *Turritopsis rubra* and *Aurelia coerulea*. a** Uniform Manifold Approximation and Projection (UMAP) visualisation of integrated cell clusters in *T. rubra* (top left) and *A. coerulea* (top right). River plots compare *T. rubra* and *A. coerulea* cluster assignments with the joint clusters, and the pie chart indicates their contributions to each cluster. Abbreviations: HA hair cell, ST/GE stem/germ cell, GL gland cell, GA gastrodermis, NEB nematoblast, NE neural cell, NEC nematocyst, EPM epidermal/muscle cell, IMM immune cell, MES mesoglea cell. **b** Comparative analyses of hair cells of *T. rubra* and *A. coerulea*. UMAP atlas of hair cells (bottom) based on integrated single-cell analyses of *T. rubra* (brown) and *A. coerulea* (aquamarine blue). Feature plots visualising the expression of genes associated with ciliary movement in hair cells.

**c** In situ hybridisation of related positive-selection genes and lost genes in *T. rubra*. Scale bar: 200 μm. **d** Volcano map (left) displaying differential gene expression between hair cells of *T. rubra* and *A. coerulea*; bar plot (right) presents the Gene Ontology analyses for DEGs. Categories involved in cilium are coloured in red. Source data are provided as a Source Data file. **e** Schematic map illustrating the hypothesis explaining statolith transportation by ciliary movement, where cilium development-related pathways are down-regulated, and the gene families kinesin (yellow) and dynein (green), which are involved in ciliary movement, are either down-regulated in the sensory organs or are not expressed in the hair cells of *T. rubra*.

RNA (siRNA against OMs and the LRR domain) during strobilation in *A. coerulea* (Fig. 3c). Following a 7 d incubation, the morphology of the polyps in the si-OM (OM siRNA interference) and si-LRR (LRR siRNA interference) groups was significantly different compared to that in the other treatment groups. In the control and siRNA negative control groups, all the polyps formed bodily segments and developed into advanced stage of strobilation (AS), with long lappets and well-developed rhopalia. In contrast, more than half of the polyps of the si-OM and si-LRR groups were still in the early stage of strobilation (ES), with undegenerate tentacles (Fig. 3d, Supplementary Fig. 13a). Subsequently, we assessed the expression of target genes using RT-qPCR and found that the si-OM and si-LRR groups exhibited significant downregulation of genes compared to the control and negative siRNA groups, validating the efficacy of the siRNA (Supplementary Fig. 13b, Supplementary Data 14) ($p < 0.001$).

### Genetic profiles of hair, neural, and muscle cells

Hair cells are located at the bottom of the statocyst chamber and may be vital for oriented movement of statolith[8,24]. To demonstrate the cellular heterogeneity among hair cells in different species, a single-cell atlas of the medusae of *T. rubra* and *A. coerulea* was generated; it contained 22,245 and 18,936 cells, respectively (Fig. 4a, Supplementary Fig. 15; Supplementary Data 15). Notably, *CFAP141*, which is absent in the *T. rubra* genome, as well as PSGs, including *TRPC4*, *CACNA1E*, *LOXHD1*, and *KIF13B*, were specifically expressed in the hair cells of *A. coerulea* (Fig. 4b). Comparative transcriptome analyses demonstrated that the hair cells of *A. coerulea* exhibited enrichment of genes associated with microtubule motor activity, cilium assembly, and the dynein complex. *DYNH8* and *DYNH1*, encoding the inner dynein arm heavy chain proteins, as well as members of the kinesin family, such as *KIF17*, *KIF4*, *KIF28P*, and *FA10*, which encode proteins involved in microtubule-based movement and motor binding, were expressed highly in *A. coerulea* hair cells (Fig. 4d, Supplementary Data 16 and 17). The up-regulated (*DNAH3, DNAH5, DNAH7, DNAH12,* and *KIF14*) and specifically expressed dynein- and kinesin-encoding genes (above-mentioned) in *A. coerulea* are likely to play a vital role in ciliogenesis, maintaining the programmed oscillation and motility of cilia, and ensuring the assembly and precise positioning of statoliths[32,33]. Together with these dynein- and kinesin-encoding genes, the down-regulation of components of the Wnt, Hippo, $Ca^{2+}$, and PI3K/Akt signalling pathways, which are involved in hair cell development in *T. rubra*, may result in the formation of non- or dysfunctional hair cells[34], contributing to the absence of statocysts in *T. rubra* (Fig. 4e, Supplementary Fig. 10).

Nerve and muscle cells are pivotal for mediating the swimming behaviour of cnidarians. Nerve cells are crucial for the coordination and transmission of signals that control movement, and muscle cells are directly involved in the physical aspects of the swimming behaviour of cnidarians swimming[35,36]. A comparison of the nerve and muscle systems of *T. rubra* and *A. coerulea* revealed distinct differences in gene expression patterns (Supplementary Figs. 16 and 17). In neural cells, specific genes associated with the regulation of muscle contraction (*CACNA1E* and *TRPC4*), sensory function (*AVIL* and *HMCN1*), and synaptic plasticity (*APBA2* and *PARD3*) were exclusively expressed in *A. coerulea*. Conversely, genes involved in signal transduction (*TRPC5*, *TRPA1*, and *SNX27*) and those related to neural function (hippocalcin and neurcalin) were highly expressed in the neural cells of *T. rubra* and *A. coerulea*, respectively (Supplementary Fig. 16, Supplementary Data 18 and 19). In the striated muscle, PSGs of *T. rubra*, including *SMTNL1*, *MYL6*, *PFN*, *RIM2*, and *CHRNN*, were specifically expressed in *A. coerulea* (Supplementary Fig. 17, Supplementary Data 20). Gene Ontology (GO) analyses revealed considerable enrichment in metal ion binding and dynein complex in *T. rubra*, whereas the myosin complex, actin cytoskeleton, and actin binding were enriched in *A. coerulea* (Supplementary Fig. 17, Supplementary Data 21).

Collectively, these findings suggest substantially different neuromuscular system profiles between the two jellyfish species, which may explain their distinct physiological and evolutionary characteristics.

### Cellular and genetic changes of forward and reverse development in *T. rubra*

To investigate the developmental pattern of *T. rubra* and reveal the cellular and genetic processes, scRNA-seq analyses was conducted on single cells sampled across the ontogenic stages of normal and reverse development, namely the medusa (Me), four-leaf structure (Ff), cyst (Cy), polyp (Po), and planula (Pl) stages. A total of 44,954 cells in 20 distinct cell clusters were assigned to nine broad cell types except one undefined cluster (Fig. 5a, b, Supplementary Data 22). Pseudo-time analyses showed that differentiation mainly involves neural, nematocyte, germ, and immune cells in Po and Pl stages, expanding to Ep/muscle cells and gastrodermis in Me and Ff stages (Fig. 5c, Supplementary Fig. 18a, b). However, in the Cy stage, stem cells stop differentiating into nematocytes (Fig. 5c). Given the pivotal role of nematocytes as remarkable cellular tools involved in predation and defence in cnidarians, we constructed a differentiation trajectory from stem cells to nematocytes during normal development and observed obvious changes in transcription factors (TFs) during nematocyte differentiation (Fig. 5c, Supplementary Fig. 18c). For example, TFs, including *TBX3*, *WNT16*, *SOXC1*, and *LMX1B* were significantly up-regulated during the intermediate phase but down-regulated at the initial and final phases; TFs, including *SOXB3*, *POU4F2*, and *FOXO* exhibited increased expression at the final phases during differentiation (Fig. 5c). It is noteworthy that during reverse development, the expression of TFs such as *E2F2* and *TBX10* significantly decreased (Supplementary Fig. 18d).

Notably, the cyst and planula stages exhibited striking similarities in terms of gene expression patterns, with enrichment of pathways related to tissue regeneration, cell proliferation, and embryonic development (Fig. 5d, f, Supplementary Data 23). Genes regulating tissue formation (*WNT4*, *WNT5A*, *WNT11A*, and *MYC*), organ development (*WNT8* and *FZD4*), and stem cell differentiation (*NANOS*, *CNIWI*, *SOXC1*, *SOXB1*, *FOXO*, and *FOXK1*) were expressed in both stages (Fig. 5f, Supplementary Data 24). Collectively, these findings suggest that the cyst stage can develop into a polyp stage, similar to the planula stage. Gene expression pattern analyses further revealed that several genes implicated in diapause[37,38], such as somatic ferritin-like protein, *CHI3L1*, *GST*, *HSP70*, and *HSP90a*, were up-regulated during reverse development (Fig. 5e, Supplementary Data 25). The expression of genes associated with cell metabolism, cell cycle regulation, and proliferation, including *CDK*, *ATP5MG*, *ATP5MO*, *COX6B1L*, and *COX6CL*, was markedly down-regulated. The distinctive characteristics of the cyst stage imply that it represents a state of diapause and may supposedly develop into the polyp stage under suitable conditions.

## Discussion

As early metazoans that transition from benthic, sessile corals to free-swimming jellyfish, cnidarians are appropriate models for studying the early adaptive evolution of animal locomotion patterns. The hydrozoan jellyfish, *T. rubra*, displays straight-swimming behaviour, ascending with contracted tentacles and descending with extended tentacles[9]. Notably, *T. rubra* lacks a statocyst for orientation and righting movements. A comparative analyses of this primitive, simple organism and closely related jellyfish species can help shed light on the early adaptive evolution of aquatic locomotor patterns. Hair cells play a pivotal role in statocyst formation. They are closely linked to otoliths, functioning as mechanoreceptors for balance and spatial orientation and potentially contribute to otolith assembly and localisation via ciliary motility[27,39]. In the present study, we identified distinct differences in the number and structure of cilia within the hair cells of the sensory organs of the *T. rubra* and *A. coerulea* harbouring statocysts.

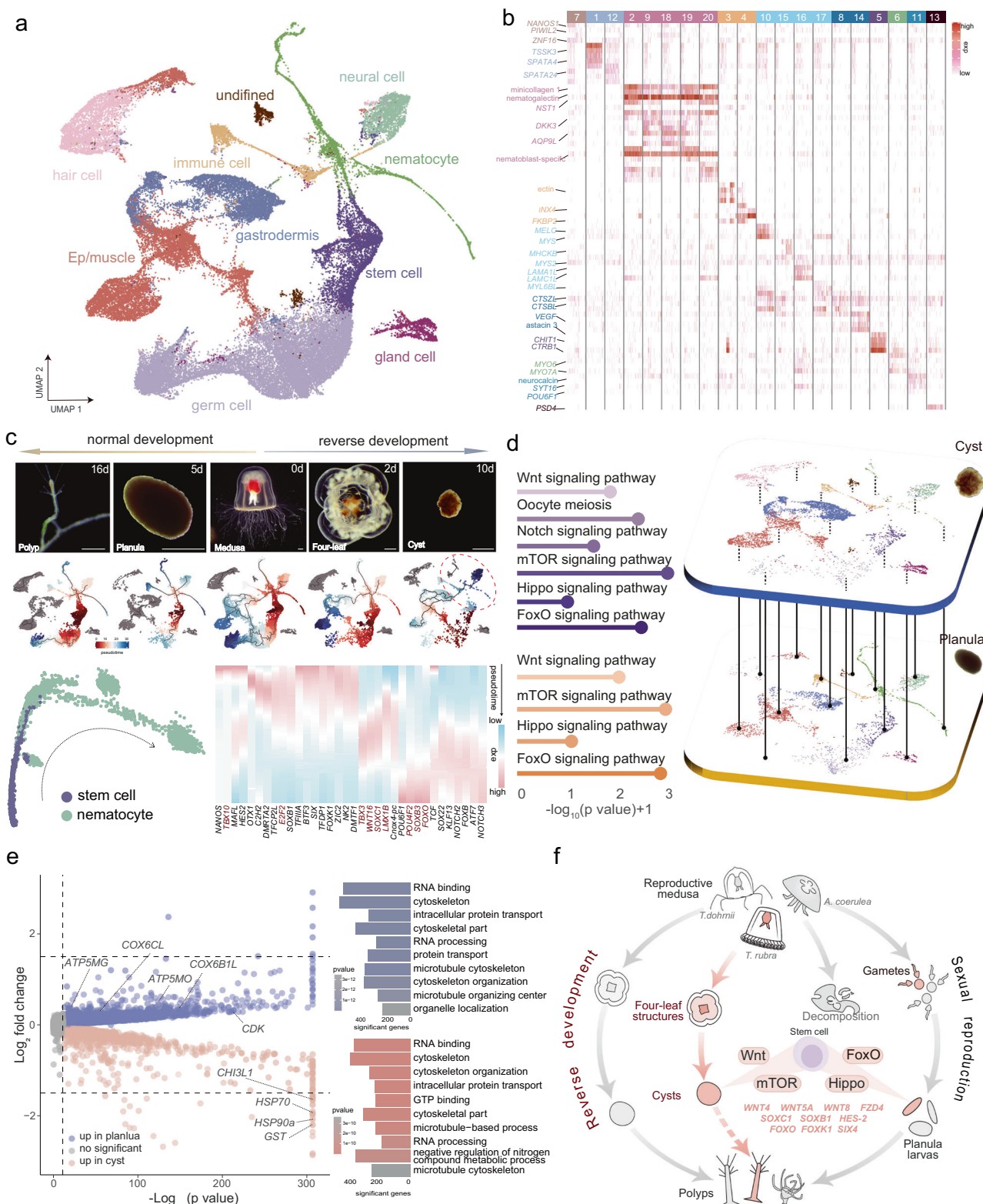

These morphological distinctions provide compelling evidence for differences in hair cells between these two species.

Ciliary motility is a prerequisite for normal otolith assembly and localisation. It requires the coordinated activity of multiple dynein motors arranged precisely along the outer doublet microtubules[28]. In *Mnemiopsis leidyi*, lithocytes are transported and added individually to the statolith through the active movement of the motile mechanoresponsive cilia surface that is powered by kinesin and dynein motors in anterograde and retrograde directions, respectively[40]. Consistently, we observed specific gene expression patterns related to dynein and kinesin families in the hair cells of *A. coerulea*, particularly within the rhopalia of three scyphozoans. In contrast, these genes were down-regulated in the sensory organ of *T. rubra*. Genes involved in microtubular and ciliary movements, motile cilium assembly, as well as

**Fig. 5 | Cell atlas of *Turritopsis rubra* in different developmental stages.**
**a** Uniform Manifold Approximation and Projection (UMAP) visualisation of the merged dataset in different stages. **b** Heatmap displaying the expression of selected marker genes per cell type. **c** Cell differentiation trajectories at different stages in *T. rubra*. Top: cell trajectory analyses of different developmental life stages based on Monocle3. Scale bar: 100 μm. Left bottom: pseudo-temporal ordering of stem cell to the nematocyte along differentiation trajectories during the normal development of *T. rubra*. The red-marked circles highlight the distinct changes in the differentiation trajectories in the podocyst stage (compared to other stages). Right bottom: expression of selected transcription factors (TFs) involved in nematocyte differentiation, ordered via Monocle2 analyses in pseudo-time. Transcription factors (TFs) changed during nematocyte differentiation are coloured in red. **d** Gene modules detected by extraction of genes from cyst and planula stages. Left lollipop chart representing the pathways of the cyst (top) and planula stages (bottom).

Right UMAP visualisation of the cell atlas between the cyst (top) and planula stages (bottom). All KEGG pathways are provided in Supplementary Data 23. **e** Volcano map (left) displaying the differential gene expression between the cyst and planula stages of *T. rubra*, and the bar plot (right) depicting the Gene Ontology analyses for gene expression across the different stages. Source data are provided as a Source Data file. **f** Part of the life cycles of *T. rubra*, *T. dohrnii*, and *Aurelia coerulea*. All species exhibit the same pattern of sexual reproduction from gametes to polyps. After reproduction, the medusa of *T. dohrnii* can develop into a polyp via reverse development rather than decomposing during normal development like jellyfish, such as *A. coerulea*. The reproductive medusa of *T. rubra* can experience a cyst-like diapause that may potentially develop into a polyp. Both the cyst stage and planula stage exhibit striking similarities in the proportion of stem cells and gene expression patterns of development, indicating that the cyst stage can develop into a polyp, similar to the planula stage.

pathways related to hair cell development were up-regulated in the rhopalia of the three scyphozoans but down-regulated in the tentacle bulbs of *T. rubra*, suggesting their significance in statocyst formation. Notably, loss or knockdown of genes (such as *CEP97* and *CFAP141*) and PSGs (such as *LOXHD1* and *USH2A*), which are required for the normal function of hair cells, results in developmental defects in otoliths and reduced swimming speed and distance in zebrafish[41,42], which, given the structural, functional, and genetic similarities of the statolith and the otolith[43], may lead to the inadequate function of motor cilia and impact statolith and statocyst formation in *T. rubra*.

Statoliths consist of calcium salts, sulphated acid mucopolysaccharides, and collagenous material[44]. Polysaccharides, such as heparan sulphate proteoglycans, chondroitin sulphate proteoglycans, and keratan sulphate proteoglycans[45], are vital for calcium carbonate-based biomineralisation[46]. The convergent amino acid substitutions in proteins encoded by two PSGs, i.e., *CHSY1* (a chondroitin sulphate proteoglycan synthase), which is critical for normal epithelial projection outgrowth, otolith growth, and tethering[47], and *GNPTAB* (GlcNAc-1-phosphotransferase), may contribute to failed statolith formation in *T. rubra*. Morpholino-based knockdown of *GNPTAB* in zebrafish results in irregular otoliths[48]. Particularly, the lost OM gene family is predicted to encode LRR-containing proteins that promote the protein-protein interactions involved in biomineralisation[46]. Specific expression of OMs in the rhopalia and their expression during in situ hybridisation downstream of the statocyst, as well as the knockdown of OMs (including LRR domains), resulted in the inability to form well-developed statocysts in the normal time frame in *A. coerulea*, suggesting that OMs play a pivotal role in statolith formation. Notably, the OM family was substantially expanded in *C. xamachana* that shows an upside-down swimming pattern, possibly because this benthic species needs more statoliths to sink underwater in an inverted position. Thus, the presence and abundance of statoliths may be associated with the differences in the swimming patterns of jellyfish. The swimming musculature of cnidarians is activated by the marginal nervous tissues[12], and the related genes likely play a role in nerve- and muscle-mediated movement regulation. These genes may also play essential roles in the evolution of the simple swimming pattern of *T. rubra*. Furthermore, genome-editing techniques such as CRISPR/Cas9 and transgenesis are not currently available for the jellyfish *T. rubra* or *A. coerulea*. These limitations collectively hinder the execution of gene functional validation experiments, impacting our in-depth understanding of these organism.

In the face of adverse environmental conditions, organisms exhibiting low motility often enter a state of diapause to adapt to unfavourable seasons and environments[49]. However, whether the reproductive medusae of *T. rubra* can develop retrogradely into polyps remains debatable. In an earlier study, *T. rubra* was found to be non-immortal, with 74% degeneration observed by day 50 of the experiment[19]. However, Miglietta proposed that Japanese *Turritopsis*, and likely *T. rubra*, could rejuvenate both before and after

reproduction, albeit at a reduced rate[18,50]. In our study, we found that the reproductive medusae of *T. rubra* can be successfully induced to develop into live cysts via starvation. This can facilitate investigation of the dynamics of transcription and cellular transition during the five crucial life stages of *T. rubra*. In the present study, we discovered that key TFs such as *LMX1B* and *TBX10*, which play critical roles in cell differentiation and organ development, are significantly down-regulated in the cyst stage of *T. rubra*[51,52]. This downregulation may potentially disrupt the differentiation of stem cells into nematocytes, likely contributing to the observed loss of predatory capabilities in *T. rubra* during the cyst stage. Moreover, high expression of diapause-associated genes (*HSP70*, *HSP90*, *GST7*, *soma ferritin*, and *CHIT3*)[37,38,53,54] and the low expression of genes associated with physiological activity (*ATP5MG*, *COX6B1*, and *CDK*)[55–57] in the cyst stage further suggest the preservation of a diapause-like state.

These gene expression patterns provide compelling evidence that the cyst stage can develop into the polyp stage. The cyst and planula stages exhibit striking similarities in terms of gene expression patterns. In our study, pathways related to tissue regeneration, cell proliferation, and embryonic development, such as the Wnt, mTOR, Hippo, and FoxO pathways, were specifically enriched in both stages. And functional genes regulating tissue formation, organ development, and stem cell differentiation were expressed in both stages. The Wnt/beta-catenin pathway has been conclusively demonstrated to play a crucial role in head regeneration in cnidarians, particularly in the freshwater polyp *Hydra*[58]. The mTOR pathway has been identified as a key regulator of organ growth in the model organism *Drosophila melanogaster*[59]. The Hippo pathway, known for its role in axis formation and morphogenesis, has been reported to exert its effects by modulating actin organisation and cellular proliferation in *Hydra*[60]. Moreover, genes involved in the maintenance of stem cell pluripotency, such as *SOXC1* and *SOXB1* expressed in the cyst stage of *T. rubra*, were substantially up-regulated during the reverse development of the 'immortal' jellyfish *T. dohrnii*[19]; the Wnt pathway and stem cell differentiation were also reported to be enriched in *T. dohrnii* cysts[61]. A comparison with the *T. dohrnii* reverse-development transcriptome analyses indicated that the cyst stage of *T. rubra* is genetically predisposed to redevelop into a polyp stage. Further, similarities in the gene expression patterns of the cyst and planula stages provided evidence that the cysts of *T. rubra* could redevelop into new polyps under suitable conditions.

In summary, we generated two high-quality, chromosome-level genome assemblies of *T. rubra* and *A. coerulea* and identified numerous PSGs and lost genes that were involved in otolith morphogenesis, ciliogenesis, and ciliary movement in *T. rubra*, indicating that these genes provided the genomic basis for statocyst formation. Transcriptomic and scRNA-seq analyses revealed that genes involved in motile cilia and hair cell function had low expression in the sensory organs and hair cells of *T. rubra*, which might explain absence of a statocyst (Supplementary Fig. 19). The dynamics of transcription at the

five developmental stages indicated that cysts can preserve themselves in during diapause and are genetically predisposed to sprout again as new polyps under suitable conditions. These adaptations partly explain the reasons underlying the absence of statocysts in *T. rubra* and its ability for reverse development. Overall, our findings provide substantial insights into the regulatory mechanisms governing these intricate developmental processes. Further exploration of the specific genes and molecular pathways implicated in these processes will deepen our understanding of the developmental biology of this species and could further provide insights into its developmental plasticity, regenerative capabilities, and adaptive responses to environmental perturbations.

## Methods

### Animal use ethics

All animal experiments were conducted in accordance with the guidelines and after obtaining the approval of the Yantai Institute of Coastal Zone Research, Chinese Academy of Sciences (2023-KJ-LL-002).

### Sample collection and nucleic acid sample preparation

In this study, two individuals of *T. rubra* and *A. coerulea* were subjected to whole-genome sequencing. Free-swimming medusae of *T. rubra* were collected from waters near Yantai and Dongying between May and August 2021 and transported back to laboratory aquariums for cultivation. The medusae were identified as *T. rubra* based on morphological observations and cytochrome C oxidase subunit I (COI) and 16S rRNA sequencing, one healthy medusa was selected for genome sequencing. The released planula larvae were collected and artificially fed to obtain asexually reproducing polyps and other medusae were used to induce the reverse development of the samples used for scRNA-seq.

The polyp strains of *A. coerulea* identified using COI and 16S sequencing were reared in laboratory cultures from planula-bearing medusae collected from waters near Yantai. One healthy polyp was selected and placed separately in a plastic bottle, fed newly hatched *A. salina* nauplii once a day with water change after 4 h, and maintained at 20 °C to promote its asexual reproduction. When approximately 100 polyps from the same individual were produced, strobilation was induced using a solution containing indomethacin (50 μmol/L). Ephyrae were produced a week later, and were collected and fed daily to ensure their development into small jellyfish. Approximately 500 small jellyfish were starved for 2–3 days before DNA extraction.

### Genome sequencing and Hi-C sequencing

For both *T. rubra* and *A. coerulea*, DNA isolated from entire medusa was used for sequencing by Novogene Technology Co. (Beijing, China) and Generead Biotechnology Co. (Beijing, China), respectively. Two paired-end libraries with a 350 bp insert size were constructed for *T. rubra* and *A. coerulea* using the Illumina Hiseq and Illumina NovaSeq, respectively. Quality-filtered reads were used for genome size estimation by employing the *K-mer* method[62]. Genome size was estimated in the following manner: Genome Size=$K_{num}/K_{depth}$. We also calculated and plotted 17-*mer* depth distributions (Supplementary Fig. 2). PacBio sequencing was performed on *T. rubra* and *A. coerulea*. Tested DNA samples were randomly broken into fragments by Covaris ultrasonic fragmentation, and large fragments of DNA were enriched and purified using magnetic beads. Thereafter, the fragmented DNA was subjected to damage and end repair; stem-loop sequencing junctions were ligated at the ends of the DNA fragments; and failed ligation fragments were removed using exonuclease. The constructed library was sequenced using PacBio Sequel.

Hi-C libraries were prepared from two adult individuals of *T. rubra* and *A. coerulea* at the previous companies. DNA was isolated from the samples, and the fixed chromatin was digested overnight using *Dpn*II.

Subsequently, the DNA was randomly divided into 300–500 bp fragments using protease digestion. Biotin-labelled DNA was captured by adsorption onto affinity beads, and the Hi-C library was prepared using end repair, A addition, splice ligation, PCR amplification, and purification, in strict accordance with the Illumina library operation procedure. All libraries were quantified using Qubit 2.0 for initial quantification and then diluted to a final concentration of 1 ng/μL for evaluation of integrity and insert size using Agilent 2100. The quality of the libraries was tested using qPCR. Thereafter, the libraries were pooled according to the effective concentration and target volume of the data to be sequenced using Illumina PE150. The Hi-C data for *T. rubra* were mapped to PacBio-based contigs using HiCup (hicup_truncater)[63]. Uniquely mapped data were used for chromosome-level scaffolding analyses. HiCUP (hicup_filter) was used for duplicate removal and quality control, and the remaining reads were used as valid interaction pairs for further assembly. The Hi-C data of *A. coerulea* was mapped and filtered using Juicer. Further details are provided in Supplementary Note 1.

### Genome assembly

The PacBio data were self-corrected, with each read being compared to the others, based on the probability of insertion, deletion, and sequencing error of the base quality. Pre-assembled reads were obtained after error correction, after which they were assembled using the overlap-layout-consensus algorithm, i.e., the overlapping relationship of the reads was leveraged for splicing to obtain the consensus sequence. The results of the previous assembly step were corrected again using Illumina paired-end reads on the Pilon software; this improved result accuracy and ended in the attainment of a high-quality consensus sequence.

Based on the Hi-C data of *T. rubra* obtained from the PacBio continuous long reads, the contigs/scaffolds sequences obtained from the assembly were mounted to the chromosome level using ALLHiC to obtain the genome at the chromosome level. For *A. coerulea*, "HiFi assembly" assembly strategy was used. The genome was assembled directly to the contig level using CCS read (also called HiFi read) by assembly software and then mounted to the chromosome level using Hi-C technology. The assembly statistics are presented in Supplementary Table 5. BUSCO assessment showed that our assembly captured 93.3% and 86.4% of the complete BUSCO values of *T. rubra* and *A. coerulea*, respectively. The sequence interaction matrices are shown in Supplementary Fig. 3, and the statistical analyses results of the chromosome assemblies are summarised in Supplementary Table 6, 7.

### Genome annotation

Sequence annotation methods can be classified into two categories, i.e., homologous sequence matching and de novo prediction. Homologous sequence matching methods were based on a database of repetitive sequences (RepBase library). Repeatmasker and repeat protein mask were used to identify sequences that were similar to known repetitive sequences. De novo prediction was performed using LTR_FINDER[64], RepeatScout[65], and RepeatModeler[66] to build a de novo repeat sequence library, which was then predicted using Repeatmasker. In addition, Tandem Repeat (TRF)[67] was used to find tandem repeats in the genome.

Protein-coding genes were annotated using a combination of de novo gene prediction programmes and homology-based methods. Further, RNA-Seq data were used to predict gene models in the genomes of *T. rubra* and *A. coerulea*. De novo prediction was performed using the Augustus[68], GlimmerHMM[69], SNAP[70], Geneid, and GenScan[71] software. For the homology-based analyses, the protein sequences of *H. vulgaris*, *C. hemisphaerica*, *R. esculentum*, *M. virulenta*, *A. aurita*, and *Pocillopora damicornis* were aligned to those of the *T. rubra* genome. Similarly, the genome sequences of *A. aurita*, *Aurelia* sp.1, *C. xamachana*, *C. quinquecirrha*, *H. vulgaris*, and *R. esculentum* were aligned to

those of *A. coerulea* via a precise spliced alignment. After combining the above prediction results with the transcriptome alignment data, EVidenceModeler[72] integration software was used to integrate the gene sets predicted by the various methods into a non-redundant, more complete gene set. Finally, using PASA[73], the EVM annotation results were corrected with the transcriptome assembly results, and UTR and variable cut information were added to obtain the final gene set. For *A. coerulea*, MAKER[74] was used to combine the gene sets. Gene functions were further annotated by searching publicly available databases, including SwissProt, TrEMBL, Kyoto Encyclopaedia of Genes and Genomes (KEGG), InterPro, GO, and NR.

An all-versus-all sequence search with an E-value cutoff of 1e-5 was performed between the predicted gene sets of the two species using Diamond (v2.0.7.145). Highly divergent gene pairs were excluded. The filtered BLASTp results were used to compute the collinear blocks using MCScanX with default parameters. Circos was used to visualise the synteny between species. More details are provided in Supplementary Note 2.

### Phylogenetic analyses

In addition to the genome sequences of *T. rubra* and *A. coerulea* obtained in the present study, we also included genome datasets obtained from NCBI [*Acropora digitifera*, *P. damicornis*, *Nematostella vectensis*, *H. vulgaris*], MGU (Marine Genomics Unit, https://groups.oist.jp/mgu) [*H. viridissima*, *M. virulenta*, *A. aurita*], published sources [*M. leidyi*[75], *Xenia* sp[76], *C. hemisphaerica*[77], *Alatina alata*[78], *C. xamachana*[78], *C. quinquecirrha*[79], *S. malayensis*[80] *N. nomurai*[81], and *R. esculentum*[82]]. Orthologues were identified using OrthoFinder (version 2.2.7) at the default settings for the 18 coelenteron species[83]. Protein sequences of one-to-one orthologues were extracted from each orthogroup using an in-house Perl script (Supplementary Software 1). Multiple alignments were generated for each of the orthogroups using MAFFT (version 7.475) with parameter '--localpair'. These were then trimmed using Gblocks 0.91b with the 'allowed gap positions' set at "With Half". A total of 1990 genes were obtained, with a tandem length of 383,088 amino acids. Individual protein alignments were concatenated using in-house Perl scripts (Supplementary Software 2). ProteinModelSelection.pl was used to deduce the best-suited substitution model for the trimmed alignment (the JTT + F + I + G4 model). For maximum likelihood analyses, the best-fit substitution model was employed as deduced by ProteinModelSelection. pl. with 1000 replicates, as implemented in RAxML (version 8.2.12). The divergence times were estimated using the Markov chain Monte Carlo (MCMC) tree in PAML with calibration. A constraint of 240 million years ago (mya) was set for the split between *Acropora* and *Nematostella*. This constraint aligns with the appearance of the first scleractinian corals in the fossil record. Other fossil calibrations were based on those previously used by Khalturin et al. [21]. Concatenated supergenes and species trees were used as input files. More details are provided in Supplementary Note 3.

### Expansion and contraction of gene families and transposable element (TE) analyses

Expansion and contraction in gene families were calculated using the CAFÉ programme (version 5.1.0) based on the birth-and-death model[84]. The parameters "-p 0.01, -r 10000, -s" were established to search the birth-and-death parameter (λ) of genes based on a Monte Carlo resampling procedure; birth-and-death parameters in gene families with a *p*-value ≤ 0.01 have been reported. Gene families without homology in the SWISS-PROT database were filtered to reduce potential false-positive expansions or contractions caused by gene prediction. The GO and KEGG terms for all proteins used in the comparative analyses were annotated using Eggnog 5.0[85]. A gene family with more than 90% of its members sharing the same annotations was considered a single functional family, and its weighting was set to one. For gene families containing sequences with multiple functional

annotations, different weighting values were assigned to each functional annotation according to the ratio of the annotation times of each term to the total annotation times of all members. The total weighting value was one for each family. More details are provided in Supplementary Note 4.

The repeats in *C. xamachana*, *C. hemisphaerica*, *M. virulenta* and *S. malayensis* were predicted as follows. Customised de novo repeat libraries were built using RepeatModeler (version 2.0.2a) and EDTA (v1.9.7). DeepTE[86] was used to classify custom repeat libraries using the built-in Metazoans database. The annotated custom repeat library and RepBase were merged to obtain a combined repeat library. Repeat-Masker (version 4.1.2) was used to predict repeats using a combined repeat library from the genome sequences. The TE information of the other jellyfish was based on previous studies.

### Gene loss

When a gene had no homologues within the *T. rubra* clade, the homologues of that gene were present in the closest sister lineage of the *T. rubra* clade, and we considered that the gene was lost in *T. rubra*[87]. One-to-one orthologous genes were extracted from each species, and multiple sequence alignments were generated. Gene loss analyses was performed using an in-house script in *R* language (Supplementary Software 3). Genes that were present in other species but absent in *T. rubra* were also manually searched and confirmed using blast the sequences in the compared genome FASTA files. More details are provided in Supplementary Note 5.

### Positive selection of genes

PSGs in the *T. rubra* lineage were tested and compared with those of all other background species. One-to-one orthologous genes were extracted from the same species set (except *H. vulgaris*) and selected for gene family expansion/contraction analyses. The coding sequences of the one-to-one orthologues were aligned using Prank v.170427 with a codon model. For codon-aligned nucleotide sequences, all mismatched and gapped codons were removed. Positive-selection analyses were conducted using the branch-site model in PAML (version 4.9j)[88]. model A (allows sites to be under positive selection; fix_omega = 0) was compared with the null model A1 (sites may evolve neutrally or under purifying selection; fix_omega = 1 and omega = 1) using a likelihood ratio test with the Codeml programme in PAML. The significance (*p* < 0.05) of the compared likelihood ratios was evaluated using χ2 tests from PAML. Then, the p.adjust function embedded in the R language was used to adjust the *p*-value using the Benjamini-Hochberg method. The significance of the false discovery rate (FDR) was set at <0.05. BS + MNM tests were also used to detect the PSGs. Orthologous genes present in other GO and KEGG enrichment analyses of PSGs were identified. More details are provided in Supplementary Note 6.

### Comparative whole-genome search for jellyfish statocyst formation-related genes

The scope of the data was manually extended to a broader range of taxa, including mammals, birds, and fish. We then scanned the genomes of a subset of statocyst formation-related genes found in PSGs, which may be important in the statocysts of seven jellyfish and the otoliths of four vertebrate species. The best-hit search in TBLASTN v2.2.30 was used to find matching reads of lost gene protein sequences in the four vertebrate genomic datasets. BLAST parameters were set to an expectation cutoff of 1e − 5, allowing a maximum number of 1000 returned sequences. The resulting sequences were BLASTed against sequences hosted on NCBI to identify the genes. The amino acid substitutions of PSGs in different species were compared using MEGA-X v10.1.8[89]. Based on AlphaFold2 and EzMol, the protein structures of the lost gene family OMs and the related PSGs were predicted.

## Scanning electron microscopy

Whole rhopalia containing the statocysts of *A. coerulea* and the tentacle bulbs of *T. rubra* were detached and chemically fixed for observation and analyses. Special care was taken to prevent mechanical damage to the tissues. Fixation was performed in 2.5% glutaraldehyde for 24–48 h at 4 °C. The samples were dehydrated in graded ethanol solutions and critical-point-dried with $CO_2$ in a Bal-Tec CPD 030 unit (Quorum K850; Quorum Technologies Ltd., UK). The dried samples were mounted on specimen stubs with a double-sided tape, gold-palladium coated with a Polaron SC500 sputter-coated unit (HITACHI MC1000, Hitachi, Japan), and viewed with a variable pressure HITACHI Regulus 8100 scanning electron microscope (HITACHI Regulus 8100, Hitachi, Japan) at an accelerating voltage of 3 kV at Wuhan Servicebio Technology Co., Ltd.

## Transcriptome analyses

We collected RNA samples from two tissues (sensory organs and bell margins (control)) of four species; these included rhopalia of three adult scyphozoan medusa (*C. quinquecirrha*, *R. esculentum*, and *A. coerulea*) and tentacle bulbs of *T. rubra* which lack statocyst, with at least four replicates for each tissue. In total, we collected 32 samples for RNA-seq. All sequencing was performed by BioMarker Technologies Company (Beijing, China). We constructed RNA-seq libraries with insert sizes of approximately 300–400 bp. The qualified library was pooled based on pre-designed target data volume and then sequenced on an Illumina sequencing platform. Raw reads from 32 different samples, in a FASTQ format, were first processed using in-house Perl scripts. Protein-coding genes for every other species were determined with BLASTp using protein sequences of four jellyfish as queries and selecting RBH pairs as orthologous genes. In total, we identified 6,428 1:1 orthologues among the four species. After mapping clean reads to respective genomes using HISAT2 (v2.0.4)[90], gene expression levels were estimated with fragments per kilo-base of exon per million fragments (FPKM values) using the Cufflinks programme for each gene in 32 samples. Pearson correlation coefficients were used to evaluate transcriptome similarity between different samples.

After combining all 32 samples, the FPKM values were transformed to a log2 scale, and one was added (log2(FPKM + 1)). Data was subjected to quantile normalisation using the R package preprocessCore, followed by Principal Component analyses[91]. DEGs that were significantly up-regulated or down-regulated between species were identified by the Wilcoxon rank-sum test using the R function, with the threshold Q-value for rejecting the null hypothesis set at 0.05. Pairwise comparisons across three combinations (*C. quinquecirrha vs. T. rubra*, *R. esculentum vs. T. rubra*, *A. coerulea vs. T. rubra*) were conducted. For each species, DEGs of the sensory organs were compared with control samples using DESeq2 and identified based on corrected *p*-values (*Q*-values) and false discovery rates with a log2 (fold change) > 1 and a *Q*-value < 0.01. GO enrichment analyses for each DEG was performed using the GOSeq R package, with a corrected $p < 0.05$ indicating significant enrichment. An interaction network of DEGs protein-protein associations was constructed using STRING with a medium confidence level (0.4). The coloured ellipses show the regrouping of the closest clustered network of all interactors using Markov Clustering MCL (inflation parameter set to 3.0). More details are provided in Supplementary Note 7.

## RNA interference (RNAi) experiment

In the RNAi experiment, two siRNAs (a mix of three design sequences) targeting OM genes and the LRR domain were designed by Ribo Bio, Guangzhou, China. The lyophilised siRNA were resuspended in RNase-free water, so that the stock concentration was 20 μM. Given that *A. coerulea* continuously resides in seawater, the siRNA soaking transfection method was employed for knockdown. Healthy *A. coerulea* polyps of similar size and from the same strain were seeded in a 12-well culture plate from the rearing tank (four polyps per well in 2 mL of 30‰ filtered artificial sea water (FASW)). To investigate the function of OM, four treatment groups were established, i.e., control group, negative control siRNA treatment group, si-OM treatment group, and si-LRR treatment group. Strobilation was induced using an induction buffer containing 25 μM 5-methoxy-2-methyl indole prepared in FASW.

For each group, 10 wells from the 12-well plate were selected for the addition of transfection complex buffer, and each group contained a total of 40 polyps. For the control group, 1 mL induction buffer was added per well. For negative control treatment group, 60 μL of 1× riboFECTTM CP Buffer, 30 μL of 20 μM negative control siRNA, 6 μL of riboFECTTM CP Reagent, and 904 μL of induction buffer were mixed per well. For the si-OM and si-LRR groups, 60 μL of 1× riboFECTTM CP Buffer, 30 μL of 20 μM siRNA stock solution, 6 μL of riboFECTTM CP Reagent, and 904 μL of induction buffer mixed combined per well. The transfection complex buffer was added to the respective treatment groups. All polyps were cultured at 19 °C for 7 d, with the transfection complex being replaced every 3 d. Subsequently, the polyps were collected for morphological observation and RT-qPCR. More details are provided in Supplementary Note 8.

## RT-qPCR

At the end of the RNAi experiment, polyps were observed, and the number of individuals at each stage of strobilation was recorded. In each group, four polyps were randomly selected to be fixed in 4% paraformaldehyde for morphological observation, and the remaining 36 polyps were chosen for RT-qPCR. mRNA was extracted using the Illustra Polyphenol Polysaccharide Plant Total RNA Extraction Kit (ZOMANBIO, Beijing, China) and cDNA was synthesised with One-Step gDNA Removal and cDNA Synthesis SuperMix kit (TransGen Biotech, Beijing, China). qPCR was performed using SYBR Green Master Mix (Thermo Fisher Scientific, Waltham, MA, USA) and the Applied Biosystems 7300 real-time PCR system (Thermo Fisher Scientific). All primers were diluted to 10 μM (Supplemental Table 17). Each analyses was repeated with three biological replicates. All samples were analysed using beta-actin as a reference gene. Expression levels were calculated using the ΔΔCt method. The data was analysed by IBM SPSS Statistics 25.

## Sc-RNA sample preparation, sequencing, and cell sorting

Planula of *T. rubra* were collected from spawning medusa, and part of planula were allowed to settle and develop in 6-well culture plates (Eppendorf) with 0.45 μm-filtered natural seawater (FSW) for approximately two weeks to collect polyps. Individual medusae found in the field were isolated into 6-well culture plates with FSW and starved to induce the four-leaf stage (5 day) and cysts (12 day). We obtained medusae of approximately 1 cm in diameter from our laboratory, where they are continuously cultured in a specialised jellyfish breeding tank.

For *T. rubra* scRNA-seq library preparation ~200 planulae, ~30 polyps, 5 four-leaf structures, 10 cysts, and 1 medusa (~1 cm in diameter) were collected and washed twice with CMFASW ($Ca^{2+}$, $Mg^{2+}$ free-artificial seawater, 25 g/L NaCl, 0.8 g/L KCl, 0.04 g/L $NaHCO_3$; pH 8.5). Immediately thereafter, samples from different life stages were separately transferred to centrifuge tubes containing 2 mL digestive buffer (3.6 mg/mL Dispase 2 and 1.25 mg/mL collagenase 1 prepared in CMFASW). Dissociation was carried out at 25 °C with occasional disruption for 20 min and was subsequently stopped by adding 8% foetal bovine serum. After dissociation, the single-cell suspension was centrifuged at 500 *g* for 5 min at 4 °C, resuspended in pre-chilled CMFASW, and then passed through a 40 μm cell strainer (FALCON, Corning, Corning, NY, USA). Cell viability was assessed using low concentrations of Calcein AM (2 mM) and Draq7™ (0.3 mM). Only cell suspensions with viability exceeding 90% were used for subsequent cell capture with the BD Rhapsody system (BD, Franklin Lakes, NJ, USA).

In this study, 10× genomics was used to capture single cells of *T. rubra* medusae and the BD Rhapsody system was used to capture single cells of *A. coerulea* medusae. FASTQ files were processed using the standard Rhapsody analyses pipeline (BD Biosciences) on Seven Bridges (https://www.sevenbridges.com) and CellRanger v1.3 software pipeline[92], following the manufacturer's guidelines. Unique molecular identifier (UMI) count matrices were imported into R (v.3.6.2) and processed using the R package Seurat (v.4.0.6) (https://satijalab.org/seurat/). To remove ambient RNA contamination and low quality, transcriptomes were filtered for each life-stage sample, with the following settings: nFeature_RNA > 200; nCount_RNA < 100; min.cells = 3 to filter out low-quality data[92]. Next, DoubletFinder package and manual screen were used to remove discrete cells[93]. To compare the differences in cell types at different stages of development, quality-filtered datasets of different life stages were merged into Seurat, and the LIGER package was used to avoid batch effects among the samples and experiments[94]. To generate cell-type clusters, the "SCTransform" function with its default parameters was used for data standardisation. Dimensionality reduction was subsequently achieved using the "RunUMAP" function, with results visualised through uniform manifold approximation and projection (UMAP). The "FindNeighbors" function (with dimensions set to 1:30) and the "FindClusters" function (at a resolution of 0.7) were also used. Following this, the "FindAllMarkers" function from Seurat was employed to identify DEGs across various clusters (Supplemental Data 13). The annotation of cell clusters was manually performed by integrating data from recognised *Aurelia* marker genes or their orthologues in other taxa (i.e., *Hydra*, *C. hemisphaerica*, and *N. vectensis*)[95–98]. More details are provided in Supplementary Note 9 and 10.

## Whole mount in situ hybridisation and imaging

The cDNA of *T. rubra* and *A. coerulea* were produced using TransScript One-Step gDNA Removal and cDNA Synthesis Super Mix (TransGen; #AT311-03). Riboprobes generated from templates produced via standard cloning and restriction digestion were synthesised using a DIG RNA labelling kit (SP6/T7) (Roche; #11175025910). Colorimetric in situ hybridisation was performed with minor modifications[99]. Briefly, in situ hybridisation samples of *T. rubra* and *A. coerulea* were relaxed in 2% MgCl$_2$ and fixed overnight at 4 °C in 4% paraformaldehyde. Hybridisation was carried out at 57 °C for 72 h. The samples were incubated with an AP-conjugated anti-digoxigenin antibody (Roche; #11093274910) and detected with NBT/BCIP substrate using a DIG Nucleic Acid Detection Kit (Roche; #11175041910). They were then transferred to 80% glycerol and imaged using LEICA DMi8 and SAPO microscopes. Further details are provided in the Supplementary Information. More details are provided in Supplementary Note 11.

## Single-cell comparative analyses

To investigate gene expression between *T. rubra* and *A. coerulea* from the aspect of single-cell transcriptome profiles, SAMAP was used to generate matrix homologous genes between *T. rubra* and *A. coerulea* (Supplementary Data 26). Next, we used the canonical correlation analyses (CCA)[100] algorithm to integrate the datasets corresponding to the medusa of *T. rubra* and *A. coerulea* with function IntegrateData and obtained the new integrated expression matrix (Supplementary Software 4). Cell clustering and dimensionality reduction of the integrated matrix were performed with the functions "FindClusters" and "RunUMAP", respectively. We used the R package river plot to compare the cluster assignments and visualise the cell-type assignments for *T. rubra* and *A. coerulea* datasets with the CCA joint clusters. Integration and analyses for the *T. rubra* and *A. coerulea* hair cells were performed in a similar manner. For each cell subset, we used the "FindAllmarker" (logFC threshold = 0.25) function in the Seurat R package to identify DEGs among different cell clusters or stages. The DEGs were then selected for mapping to the GO and KEGG databases using the corresponding genome annotation information. The R packages gene set variation analyses (GSVA) and ClusterProfiler were used to identify the GO and KEGG terms between samples and cell clusters.

## Pseudo-time analyses

The R package Monocle 3[101] was utilised to infer cell differentiation trajectories across different stages and all cell clusters. The R package Monocle 2[102] was employed to construct the trajectories for stem cell differentiation into nematocytes and to analyse the dynamics of gene expression during the normal development of *T. rubra*. Furthermore, the R package cytoTRACE[103] was used to verify and identify the starting points of differentiation and differentiation potential across various stages in *T. rubra*.

## Statistics and reproducibility

For each sample, scanning electron microscopy was conducted twice. In the RNAi experiment, four polyps of each group were collected for morphological observation. The micrographs of the in situ experiment were captured twice. In the in situ hybridisation experiment, eight replicates were set up for each gene in both species. Micrographs of each developmental stage of *T. rubra* were taken three times. No statistical method was used to predetermine sample size. No data were excluded from the analyses.

## Reporting summary

Further information on research design is available in the Nature Portfolio Reporting Summary linked to this article.

## Data availability

The whole-genome assemblies of *T. rubra* and *A. coerulea* have been deposited in the NCBI database under accession code PRJNA1005405. The raw reads of the RNA-seq of the four jellyfish have been deposited in the NCBI database under accession code PRJNA1010405. The raw reads of the single-cell RNA-seq of *T. rubra* and *A. coerulea* have been deposited in the NCBI database under accession code PRJNA1045549. Source data are provided with this paper.

## Code availability

The codes used in the study are provided in the Supplementary Software files 1–4, and are available in our lab's GitHub repository (https://github.com/Changhao051/Turritopsis-rubra).

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

## Acknowledgements

This research was supported by the National Natural Science Foundation of China (U2106208 to Z.D., 42425004 to Q.L., 41876138 to Z.D.), National Science & Technology Fundamental Resources Investigation Program of China (2022FY100600 to Z.D.), National Key Research and Development Program of China (2023YFC3108200 to Z.D., 2022YFC3102403 to Y.Liu), Strategic Priority Research Program of the Chinese Academy of Sciences (XDA23050301 to Z.D.), Taishan Scholars Program (tsqn202211263 to Z.D.), and Seed Project of Yantai Institute of Coastal Zone Research, Chinese Academy of Sciences (YICE351010101 to J.Z.).

## Author contributions

Q.L., Z.D. and J.Z. conceived the project. Q.L. and Z.D. supervised the study. L.W. and T.S. collected the samples. F.W., H.Y., Y.Liu, M.Q., Z.D. and Q.L. performed genome analyses. F.W., Y.Li, T.S. and Y.M. performed transcriptomic analyses. F.W. and Z.D. performed scanning electron microscopy analyses. Z.D., Y.Li, K.S. and K.C. performed single-cell RNA sequencing analyses. S.P. performed the in situ hybridisation. Z.D., F.W., Y.Li, Q.L. and Y.Liu wrote the manuscript with input from all other authors. All authors reviewed and contributed to the final manuscript.

## Competing interests

The authors declare no competing interests.
