## [Peer Review File · Nature Communications]

Genomic and single-cell analyses reveal genetic signatures of swimming pattern and diapause strategy in jellyfishREVIEWER COMMENTS

Reviewer #1 (Remarks to the Author):

The manuscript reports two chromosome level genome assemblies of the Medusozoa *Turritopsis rubra* (a hydrozoan) and *Aurelia coerulea* (a scyphozoan). In addition there is a single cell dataset of different developmental stages of *Turritopsis* and some additional single cell data from *Aurelia* hair cells. The authors use these data to address questions of statocyst formation, swimming strategies and reverse development (in *Turritopsis*). Although the data are interesting, there are serious problems with the manuscript that prevent me being enthusiastic about it.

The genomes are of high quality (i.e. chromosomal level), but they are not adequately placed in context. Another *Turritopsis rubra* genome has been published (Pascual-Torner et al. PNAS, 2022). Clearly this is relevant. Although this paper is cited the genome is not mentioned. No comparison is done to show that the assembly of the current work is better. The Pascual-Torner paper also includes a *T.dohrnii* genome and there is a further published *T.dohrnii* genome from Hasegawa et al. 2023 of somewhat better quality. The Hasegawa paper is not cited. These data should clearly be reviewed in the introduction, and as much of the present manuscript concerns patterns of evolution in *T.rubra*, including gene loss and positive selection, it seems likely that they would be useful for testing and validation purposes. Similarly prior work on *Aurelia* is not cited.

Beyond this, the analyses are poorly and confusingly presented. For instance, in the section "Genetic basis of statocyst and movement regulation [...]", there is no objective description prior knowledge of what genes are known to be involved in cnidarian statocyst formation. Instead a list of positively selected genes (PSG) is asserted to be relevant (CHSY1, LOXHD1, USH2A, PLS1) and referred to supplementary data tables with GO term enrichments and p-values for positive selection (but no dN/dS ratios). The methods describing these analyses are split between the main manuscript and the supplementary information making them difficult to follow, and establish exactly what has been done. Further, I am not sure that the table numbering is correct (e.g. the spreadsheet labelled internally in row 1 as Supplementary Data 5 seemd to be referred to in the manuscript as Supplementary 6). Genes are referred to by human names in the text, but not in the spreadsheet. E.g. in the key table of PSGs labelled as supplementary table 4, I can find CHSY1, but not LOXHD1, USH2A or PLS1.

Given the centrality of the PSG analysis, all alignments should be made available, as misalignment is likely to lead to artefactually inflated dN/dS values. This is not addressed at all.

All the core analyses involve testing large numbers of genes and then focussing on particular candidates, with no compelling case being argued for their genuine involvement in the relevant processes. Cited literature used as support is from unrelated species. Rather, it looks as though genes have been picked from lists based on keywords. The p-values given in the tables are generally quite high (i.e. less significant), and it is not clear that all multiple testing adjustments have been carried out. Genes appear randomly in the discussion (e.g. "convergent amino acid substitutions" in CHSY1 and GNPTAB, with no other mention of GNPTAB that I can find).

I think this manuscript demands a great deal of further work to turn it into either a useful presentation of the data generated, or an unbiased investigation in statocyst and/or behavioural evolution.

A small selection of more specific comments:

introduction

L.75: "primitive species" - problematic - all extant species are equally evolved and non can be regarded as primitive. Find another phrasing.

L.76-77: "rendering it an ideal model" - not really, as striated muscle has been argued to be convergent in medusae (Steinmetz et al. Nature, 2012) and doubtful too about the nervous system as the swimming behaviour is unique to the medusae within Cnidaria - unlikely to be plesiomorphic.

L.80-84: "ambush predators" etc. These terms need further explanation as they are unlikely to be familiar to non-specialists. The argument that they have low motility and therefore find it hard to escape and therefore rejuvenation is a survival strategy is purely speculative and unconvincing.

results

L.117-120 phylogenetic reconstruction: please give some summary numbers of the final alignment used for this analysis - no. of genes, no of positions, missing data. I can't see this information here, or in the methods.

L.124-127: "may coincide with the emergence of the statocyst and the ability of jellyfish to swim freely" - apropos of a Clytia/Turritopsis clade being compared to a scyphozoan clade, what is the relevance of this remark?

L.136 Figure 2a - it is not clear how the zoomed regions relate to the tentacle bulb / statocyst. Can they be boxed? The paragraph talks about "a small, stone-like object" but it is not clear if this is pictured or not (I don't think it is).

genome annotation methods

L.489-492: the choice of cnidarian species for homology-based analysis appears inconsistent e.g. Turritopsis includes an anthozoan (*P.damicornis*) and a box jellyfish but *Aurelia* does not. Why?

L.498: why was MAKER used for *A.coerulea* but not *T.rubra*?

Supplementary methods, genome annotation L.64: *embryophyta_odb10* for a BUSCO database is clearly inappropriate as this is a plant dataset.

Reviewer #2 (Remarks to the Author):

Dong et al. have sequenced and compared two high-quality chromosome-level genomes of jellyfish with other cnidarian genomes, focusing on the genes related to the statocysts and hair cells of species with different swimming patterns during the evolution of jellyfish locomotion. They further employed transcriptomic and scRNA-seq analyses and found that genes involved in motile cilia and hair cell function were lower expressed in the sensory organs and hair cells of *Turritopsis rubra*.

These results revealed the genetic basis for the absence of jellyfish balance organs, and the discovery of a potential gene family “OM” is quite interesting from a developmental and genomics evolutionary perspective. Generally, the manuscript is well-crafted, demonstrating considerable efforts to investigate the genetic foundation of jellyfish locomotor characteristics and survival strategies. The data presented in this study would enhance our understanding of the evolutionary lineages of movement and adaptation within the animal kingdom. I have a few clarifying questions and some comments aimed at enhancing the manuscript.

This study mainly focuses on the genetic basis of the absence of statocyst in *Turritopsis rubra*, however the title refers to the “specialised swimming pattern”, is there a direct/or indirect link between statocyst and jellyfish swimming pattern? The authors should give more explanation and provide some references.

This study identified a number of nerve and muscle related genes in the comparative genomes and also analyzed and discussed in the single cell comparisons of nerve and muscle cells, why did you study them? A brief explanation of their selection will shed light on their relevance to the research objectives.

Single-cell transcriptomic analyses revealed the enrichment of diapause formation-related genes in the cyst during the reverse development of *Turritopsis rubra*, were any genomic comparisons made to support this at the genomic level? In addition, you mentioned to the fact that there have been comparative genomic analyses to study the reverse development of two *Turritopsis*. Have you also made genome comparisons in your study to obtain similar or opposite conclusions? This issue should also be discussed.

Some methods need improvements or clarification. For example, positive selection analyses were identified solely using the branch-site model in PAML, however, multinucleotide mutations can cause false inferences of lineage-specific positive selection. So, BS + MNM test were suggested to detect positive selection. Also, it is unclear how reliable of the lost genes found in this study.

References are missing for some softwares, e.g., The amino acid substitutions of PSGs in different species were compared using MEGA-X v10.1.8. CellRanger v1.3 software pipeline, following the manufacturer’s guidelines.

Line 66-68: add specific information on the calibration points.

Line 186: I think “CEP141” should be “CFAP141” according to the pictures and discussion?

Line 203: again, CFAP141.

Line 252: the phrase "substantial enrichment" is somewhat vague.

Line 270: the use of "changed" is preferred over "altered" for a clearer implication of variance in the study's context.

Line 286-288: add the relevant literature to support your claims.

Line 371-372: given the complexity of Wnt, mTOR, Hippo, and FoxO signaling pathways, please provide specific examples where these pathways have been implicated in tissue regeneration or dormancy germination across different species.

Line 420: polyps.

Line 420: "Ephyra" should be "ephyrae".

Line 490: the correct form of Aurelia sp1 should be Aurelia sp.1.

Line 507: T. rubra

Line 543: provide version information for DeppTE.

Line 624: $1E-5$.

Line 646: are the cells retained on or passed through a 40- μ m strainer?

Line 665: ensure consistency in the reference to functions. It is helpful to have them in quotes, and might also help to add the word "functions" after "FindNeighbors" and "FindClusters".

Line 647: How to determine it?

Line 655-657: why you setting this criteria? Is there any citations?

Line 710: Verify the accession code PRJNA1014582, as the datasets are currently not accessible.

The indicator lines in Figure 1b are not in the correct position, check and correct them.

Reviewer #3 (Remarks to the Author):

In this manuscript, Dong and colleagues sequenced one new hydrozoan and one new scyphozoan genomes, analysed the expansion and loss of genes, and carried out single-cell transcriptomic analyses of five developmental stages. Despite the field of animal genomics has rapidly advanced in last two years and obtaining new genomic resources have different weight nowadays, I think this manuscript has intriguing results in its gene loss and single-cell data analyses.

Regarding the gene loss analyses, it is generally well known to be difficult to prove in the field, and the authors should provide multiple lines of evidence to support their claims. For instances, are these genes also not contained in the transcriptomes in respective and related species? How about genomes of other related species that are not included in the current analyses, can the authors look into them and ensure they are also not there? Other additional analyses could also be included such as syntenic and gene pathway (than individual gene) analyses. These will make the claim much more solid.

Reviewer #4 (Remarks to the Author):

Dong et al. conducted a study investigating the comparative genomic and single-cell transcriptomic analyses of two jellyfish species, namely *Turritopsis rubra* and *Aurelia coerulea*. They performed de novo assemblies of two reference genome and identified several candelatae genes associated with swimming patterns, specifically PSGs. Additionally, they performed comparative cellular analyses across two species and five developmental stages of *Turritopsis*

rubra, leading to the identification of multiple candidate different expressed genes in hair cells (e.g., PSGs) that may be involved in swimming patterns, and the diapause formation-related genes in the cyst during the reverse development of *Turritopsis rubra*, respectively. While this manuscript presents an important resource for studying the movement patterns and shape of jellyfish, there are a few areas that require further clarification and more robust analysis. Here are my major comments:

1. In the section of sequencing and genome assembly, it is mentioned that *A. coerulea* was directly sequenced using Pacbio Hifi CCS technology. However, it was not clearly stated which sequencing technology was used for *Turritopsis rubra*. Was Pacbio CLR or Hifi CCS employed for this species? It is also important to provide information about the assembly software or pipelines used, including version numbers and parameters. Additionally, an explanation is needed for why the Hi-C data of the two species were processed using different pipelines. It is crucial to evaluate any potential issues that may arise from using different sequencing technologies or assembly pipelines. For instance, it is worth investigating why the contig N50 of *Aurelia coerulea* assembly is notably longer than that of *Turritopsis rubra*, and why the complete BUSCOs of *Aurelia coerulea* exhibit relatively lower completeness and higher fragmentation compared to *Turritopsis rubra*.

2. The author employed different single-cell sequencing platforms, namely 10X Genomics and BD Rhapsody, to generate single-cell transcriptomic data for *T. rubra* and *A. coerulea*. However, the potential batch effect resulting from using these different technologies in the comparative cellular analysis should be thoroughly evaluated. Additionally, the author mentioned the dynamic changes in cell composition proportions across different developmental stages of *T. rubra*, which necessitates increased biological replication due to potential bias introduced by two scRNA-seq technologies and individual differences. These factors can significantly impact cell composition and consequently influence the conclusions drawn from this study. Furthermore, accurate classification of cell types is a critical and challenging task for two non-model species. Therefore, the author should provide details regarding their approaches to defining cell types and assess the annotation accuracy of cell types in both species across the five different developmental periods of *T. rubra*.

Minor comments:

1. Figure 2a is positioned before Figure 1d in the text and requires adjustment in terms of layout. And the two ordinate labels in Figure 1d are placed closely together, potentially causing confusion. It is recommended to adjust and differentiate them.
2. The scale in Figure 2a appears too small and blurry, making it difficult to view clearly.
3. In lines 140-143, the authors identified a total of 548 PSGs. Considering the relatively large divergence time of these species, the authors should carefully examine the alignment results of

selective amino acid loci and their upstream and downstream sequences in these PSGs. This is important to mitigate the risk of higher false positives in PSG identification.

4. In lines 186-187, the authors mentioned in the manuscript that "the loss of CFAP141 and cep97 in *T. Rubra* may also result in the absence of Statoliths," while "CFAP141" is marked in the figure. This appears to be a spelling error that needs correction.

5. It is important to assess the loss genes shown in Figure 3a, including CFAP141, CEP97, KCNK1, NSMF, and MAOA, due to the fact that they exhibit only 1-2 gene copies in other comparative species. The evaluation should consider potential sequencing bias, assembly integrity, gene annotation integrity, or use PCR experimental data.

6. In lines 193-195 and Figure 3b, considering the substantial divergence time between the used species (from jellyfish to mammals), it is necessary to examine the alignment results of upstream and downstream amino acid sequences of these conserved sites.

7. The in situ hybridization results in Figure 4c appear to be fuzzy and unclear. It is recommended to replace this image with a high-definition picture to improve visibility.

8. In Supplementary Figure 5, the overlapping and illegible names of the samples needs to be addressed for improved readability.

9. In Supplementary Figure 10a, the color of the cell type numbers does not correspond to the color of the cells themselves. This discrepancy should be rectified for consistency.

10. In Supplementary Figure 10b, certain marked genes are not unique to specific cells. For example, the gene MEIG1 is marked for both cells 8 and 2.

11. It is suggested to use a semicolon to indicate numerical values in the supplementary tables, with two decimal places retained. For instance, Supplementary Tables 11 and 14 should follow this formatting recommendation.

Point-by-point response to reviewer's comments

Reviewer #1 (Remarks to the Author):

The manuscript reports two chromosome level genome assemblies of the Medusozoa *Turritopsis rubra* (a hydrozoan) and *Aurelia coerulea* (a scyphozoan). In addition there is a single cell dataset of different developmental stages of *Turritopsis* and some additional single cell data from *Aurelia* hair cells. The authors use these data to address questions of statocyst formation, swimming strategies and reverse development (in *Turritopsis*). Although the data are interesting, there are serious problems with the manuscript that prevent me being enthusiastic about it.

Response: Thank you for your comments and critique on our manuscript. We appreciate your insightful feedback, which has greatly helped to improve our manuscript. We have addressed all your specific comments and provided a point-by-point response below.

Questions for clarifications:

Comment 1: The genomes are of high quality (i.e. chromosomal level), but they are not adequately placed in context. Another *Turritopsis rubra* genome has been published (Pascual-Torner et al. PNAS, 2022). Clearly this is relevant. Although this paper is cited the genome is not mentioned. No comparison is done to show that the assembly of the current work is better. The Pascual-Torner paper also includes a *T.dohrnii* genome and there is a further published *T.dohrnii* genome from Hasegawa et al. 2023 of somewhat better quality. The Hasegawa paper is not cited. These data should clearly be reviewed in the introduction, and as much of the present manuscript concerns patterns of evolution in *T.rubra*, including gene loss and positive selection, it seems likely that they would be useful for testing and validation purposes. Similarly prior work on *Aurelia* is not cited.

Response: Thank you very much for your comments. We agree that a comparison of our assemblies with those in previous studies is necessary.

1. We have cited the paper related to *Turritopsis* and *Aurelia* genome in our manuscript (Line 99-101). According to your suggestion, we have compared our two genomes with the currently published information for the three *Turritopsis* genomes (Pascual-Torner et al, 2022; Hasegawa et al, 2023) and three *Aurelia* genome assemblies (Gold et al, 2019; Khalturin et al, 2019), and have added this content to the Results section of the revised manuscript (Line 124-127) and Supplementary Information (Supplementary Table 13). Comparative result shows that all previous genome assemblies were at the scaffold or contig level, whereas our genomes represent chromosome-level assemblies for *Turritopsis* and *Aurelia*.

Supplementary Table 13: Comparison of genome assemblies of *Turritopsis* and *Aurelia* with published genomic statistics.

Species	T. rubra	T. rubra	T. dohrn ii	T. dohrn dohrnii	A. coerulea	Aurelia sp.1	A. aurita (Atlantic)	A. aurita (Pacific)
Assembly size (bp)	266.86	210.00	390.00	435.92	566.06	713.00	377.00	429.00
Contig num	762	53,262	68,044	891	42	67,005	170,088	213,756
contig N50 (bp)	1,187,606	3,457	7,666	747,194	22,395,985	20k	2,627	2,665
Scaffold num	329	9,508	74,829	—	22	16,793	2,710	7,744
scaffold N50 (bp)	16,943,197	71,856	10,419	—	25,260,120	124K	1.04M	0.2M
GC Content (%)	34.26	34.00	34.50	34.70	37.34	32.60	37.10	37.60
TE rate (%)	47.43	39.45	50.78	60.35	73.14	49.50	44.67	44.03
Complete BUSCOs (%)	92.40	88.78	78.88	90.40	86.40	—	79.80	78.10
Gene num	18,746	9,324	17,468	23,314	32,035	29,964	28,625	30,166
Genome coverage	200×	96×	95×	219.5×	120×	—	90×	90×
Assembly level	chromosome	scaffold	scaffold	contig	chromosome	scaffold	scaffold	scaffold
Reference	This study	Pascual-Torner et al., 2022	Hasegawa et al., 2023	This study	Gold et al., 2019		Khalturin et al., 2019	

2. In addition, we strongly agree with your point about testing using these published genomes to verify that immediate homologous genes are conserved between different species in these same genera in order to increase the credibility of our result. Therefore, we first downloaded the other three published genome sequences of the genus *Turritopsis* from NCBI (*Turritopsis_dohrnii*_S001: GCA_025167195.1, *Turritopsis_dohrnii*_TUR: GCA_027922465.1, and *Turritopsis_rubra*_GCA_025167575.1). Since NCBI did not provide the gene prediction gff3 files, we downloaded the raw sequencing data of the transcriptomes (72 samples for *T. dohrnii*, and our transcriptomic data for *T. rubra*) and used transcriptome prediction (Hisat2, StringTie, Scallop2, TrandDeCoder, PASA2) and homology prediction (GeMoMa) to predict the gene models. EvidenceModeler (EVM) software was used to integrate the final genes.

Since the number of genes predicted by *Turritopsis_dohrnii*_TUR was only half of that in our study, to ensure the accuracy of the results, we added only two genomes of *T. dohrnii* to re-run a version of the positive selection and gene loss analyses for testing and validation, and the results are also uploaded separately as support for the responses.

For the positive selection, fewer PSGs were obtained when three *Turritopsis* were used as the foreground (*Turritopsis_rubra*-as-foreground.PSG.xls), possibly due to interspecific differences resulting in different third codons for the same amino acid, which prevented them from being screened. When two *T. dohrnii* were used as foreground, most of the selected genes obtained overlapped with the results of our original analysis (*Turritopsis_dohrnii*-as-foreground.PSG.xls). For gene loss analysis, a total of 126 gene families were lost in the genomes of three *Turritopsis*, and key candidate genes were absent in all genomes, indicating the confidence of our gene loss results and the results are also uploaded separately as support for the responses (*Geneloss.AddSpe2.xls*).

3. To further increase the confidence of our analyses, we also used these published genomes to test and validate our key genes on positive selection and gene deletion.

Key genes of positive selection: the protein sequences of these three species were compared to the positively selected genes from our study, the optimal comparison was added to the homologous gene sequence set, the homologous gene sequences were compared, and the comparison was not de-gapped to ensure that the sequences upstream and downstream of the positively selected locus were available. The upstream-downstream comparison of the key genes listed in Figure 3b is shown in the flowing Figure A. The selected amino acid sites were almost all conserved in the genus *Turritopsis*, except *CDH23*, which was conserved in *T. dohrnii* and *T. rubra*, respectively, but the corresponding amino acid sequences were also different when compared to jellyfish possessing statocyst, providing high confidence in our results.

Figure A. The upstream and downstream sequences of the selective amino acid loci of the key PSGs.

Key genes of gene loss: The other three genome sequences of the genus *Turritopsis* were also used to validate our gene loss results in Figure 3a. Genomic DNA sequences were compared using the tblastn software to rule out the possibility of gene loss due to gene prediction. The sequence comparison result files and the distribution of blast score values are provided in file GeneLoss_Blast.xlsx, and "Homolog" is the blast score value for comparison between homologous sequences. The results indicated that these loss genes are indeed absent from *T. rubra* and related species such as *T. dohrnii*.

Figure B. Sequence matching and distribution of blast score values.

In addition, the BS + MNM test was also performed to detect the PSGs as suggested, and the text and figures of the manuscript have been updated based on the newly generated data. However, since our article only focuses on *T. rubra*, and our subsequent transcriptome and single-cell transcriptome analyses used *T. rubra* as the sample due to sampling difficulties, the original version of the species selected for analyses is still used in the manuscript, with the addition of the three *Turritopsis* genomes in the response to validate our results.

References:

- Pascual-Torner M, Carrero D, Pérez-Silva JG, López-Otín C. Comparative genomics of mortal and immortal cnidarians unveils novel keys behind rejuvenation. *Proc Natl Acad Sci U S A* **119**, e2118763119 (2022).
- Hasegawa Y, Watanabe T, Otsuka R, Toné S, Kubota S, Hirakawa H. Genome assembly and transcriptomic analyses of the repeatedly rejuvenating jellyfish *Turritopsis dohrnii*. *DNA Res* **30**, dsac047 (2023).

Gold DA, *et al.* The genome of the jellyfish *Aurelia* and the evolution of animal complexity. *Nat Ecol Evol* **3**, 96–104 (2019).

Khalturin K, *et al.* Medusozoan genomes inform the evolution of the jellyfish body plan. *Nat Ecol Evol* **3**, 811–822 (2019).

Comment 2: Beyond this, the analyses are poorly and confusingly presented. For instance, in the section "Genetic basis of statocyst and movement regulation [...]", there is no objective description prior knowledge of what genes are known to be involved in cnidarian statocyst formation. Instead a list of positively selected genes (PSG) is asserted to be relevant (CHSY1, LOXHD1, USH2A, PLS1) and referred to supplementary data tables with GO term enrichments and p-values for positive selection (but no dN/dS ratios). The methods describing these analyses are split between the main manuscript and the supplementary information making them difficult to follow, and establish exactly what has been done. Further, I am not sure that the table numbering is correct (e.g. the spreadsheet labelled internally in row 1 as Supplementary Data 5 seemd to be referred to in the manuscript as Supplementary 6). Genes are referred to by human names in the text, but not in the spreadsheet. E.g. in the key table of PSGs labelled as supplementary table 4, I can find CHSY1, but not LOXHD1, USH2A or PLS1.

Response: We are sorry that the presentation of our analyses was confusing, and your comments are of great value for improving the logic and readability of our manuscript. Please allow us to respond to each of your comments as follows:

1. We are acutely aware of the importance of adding relevant background studies to the presentation of the analyses. In the area of sensory organ evolution and developmental research on cnidarians, some homeobox genes encoding the homeodomain transcription factors Six and Pou have been reported to be expressed in the statocysts of *Craspedacusta sowerbyi* (Hroudova *et al.*, 2012), and both Fox and Hox have been reported to be expressed in the ciliated mechanosensory cells in the statocysts of the *Clytia hemisphaerica* (Chevalier *et al.*, 2006). However, using BLAST (v2.2.30) software and genomic Fasta files, we manually searched for and confirmed that these reported genes also exist in the genomes of *Turritopsis rubra* and *Hydra vulgaris*, which have no statocysts, suggesting that they are not specifically associated with cnidarian statocyst formation. Since there are few studies related to the genetic mechanism of statocyst formation in cnidarians, and given the structural, functional and genetic homology of statoliths with vertebrate otoliths (O'brien *et al.*, 2003; Fritzschn and Straka, 2014), the listed genes are direct homologues that are present and conserved in cnidarians and have been verified to be associated with otolith formation in a variety of model organisms. Therefore, we speculate that these genes may be related to the statocysts of jellyfish and we emphasise this in the discussion section of the revised manuscript to reduce confusion (Line 351-352). In addition, the missing dN/dS file was provided in the resubmitted Supplementary Data (Supplementary Data 6).

2. We apologise for any confusion caused by the description of our methods section. The "Positive selection of genes" included in the Supplementary Information is complementary to the results of the two sections "Positive selection of genes (PSGs)" and "Comparative whole-genome search for jellyfish statocyst formation-related

genes”, which are described in the manuscript's Methods section in two separate subsections. We have reorganised these two parts of the methods in the revised manuscript to give the reader a clearer understanding of what we have done (Line 622).

3. Thank you for pointing out the numbering errors in our supplementary forms. We have double checked the corresponding form numbers to make sure they are correct in the revised files.

4. The purpose of referring to genes by their human names is to present genes in a clearer and simpler way and to improve the readability of the manuscript. Thank you for your comments, which made us realise that inconsistencies between the main text and the supplementary data could be confusing for the reader. To make the manuscript more rigorous, we have added “Symbol” to the corresponding Supplementary Data (Supplementary Data 5).

References:

Hroudova M, *et al.* Diversity, phylogeny and expression patterns of Pou and Six homeodomain transcription factors in hydrozoan jellyfish *Craspedacusta sowerbyi*. *PLoS One* **7**, e36420 (2012).

Chevalier S, Martin A, Leclère L, Amiel A, Houliston E. Polarised expression of *FoxB* and *FoxQ2* genes during development of the hydrozoan *Clytia hemisphaerica*. *Dev Genes Evol* **216**, 709-720 (2006).

Fritzscht B., Straka H. Evolution of vertebrate mechanosensory hair cells and inner ears: toward identifying stimuli that select mutation driven altered morphologies. *J Comp Physiol A* **200**, 5-18 (2014).

Comment 3: Given the centrality of the PSG analysis, all alignments should be made available, as misalignment is likely to lead to artefactually inflated dN/dS values. This is not addressed at all.

Response: Thank you for your insightful comment. The software we used for PSG analysis, Codeml, has its own step of de-gapping. Coding sequences of the one-to-one orthologues were aligned using Prank, and the protein alignment was conducted using the amino acid sequences translated from the aligned coding sequences, such that the protein sequences were themselves aligned. In addition, we have supplemented the Methods section of the revised manuscript with a description of our treatment of gaps to ensure that all alignments are usable (Line 610-611).

Comment 4: All the core analyses involve testing large numbers of genes and then focussing on particular candidates, with no compelling case being argued for their genuine involvement in the relevant processes. Cited literature used as support is from unrelated species. Rather, it looks as though genes have been picked from lists based on keywords. The p-values given in the tables are generally quite high (i.e. less significant), and it is not clear that all multiple testing adjustments have been carried out. Genes appear randomly in the discussion (e.g. "convergent amino acid substitutions" in CHSY1 and GNPTAB, with no other mention of GNPTAB that I can find). I think this manuscript demands a great deal of further work to turn it into either

a useful presentation of the data generated, or an unbiased investigation in statocyst and/or behavioural evolution.

Response: Thank you for your insightful comments regarding the presentation of positive selection results. Your comments are of great value to improve our manuscript. We have responded to each of your comments as follows:

1. These candidate genes, although rarely reported in jellyfish, have been functionally validated in model species such as zebrafish and mice, as cited in the text, and have similar functions in several species. Thus, based on the similarity of protein sequences and predicted structures, we suggest that they may have similar functions in jellyfish as well. However, to increase the confidence of our results, we also performed RNA interference experiments during the strobilation stage of *A. coerulea*, when the statocyst begins to form, to explore the possible biological functions of the candidate OM genes. The methods and results were added to the revised manuscript (Line 218-232, Line 680-716).

2. The multiple testing adjustments have been carried out according to your suggestion, and the results are provided in the resubmitted Supplementary Data 5. These genes were selected on the basis of their possible biological functions and the p-value after multiple testing was still less than 0.05, so we suggest that they could be used as candidate genes.

3. Thank you very much for pointing out the random occurrence of genes in the discussion, we apologise for the omission during the writing of the manuscript and appreciate the opportunity to clarify. GNPTAB is present in Figure 2 and Extended Data Figure 4 in the results section, but the lack of emphasis in the text section led to confusion, so we have added information to improve the logic of the manuscript (Line 161).

4. We have carried out more work based on your comments, including **comparative analyses of published genomes, multiple testing of PSGs, and biofunctional probing of candidate genes**, which have made our study more convincing. We have also revised potentially confusing descriptions in the manuscript to effectively present the results and to improve the rigour and readability of the manuscript.

A small selection of more specific comments:

introduction

Comment 6: L.75: "primitive species" - problematic - all extant species are equally evolved and non can be regarded as primitive. Find another phrasing.

Response: Thank you for your valuable suggestion to use more precise terminology in our manuscript. We have revised the manuscript to replace the term "primitive species" with "early branching species" to ensure accuracy and clarity in our evolutionary descriptions (Line 79).

Comment 7: L.76-77: "rendering it an ideal model" - not really, as striated muscle has been argued to be convergent in medusae (Steinmetz et al. Nature, 2012) and doubtful

too about the nervous system as the swimming behaviour is unique to the medusae within Cnidaria - unlikely to be plesiomorphic.

Response: Thank you for your insightful suggestion. We have rewritten the sentence in the revised manuscript to make our presentation more accurate (Line 79-82).

Comment 8: L.80-84: "ambush predators" etc. These terms need further explanation as they are unlikely to be familiar to non-specialists. The argument that they have low motility and therefore find it hard to escape and therefore rejuvenation is a survival strategy is purely speculative and unconvincing.

Response: Thank you for pointing out that overly specialised terms are unfamiliar to non-specialists. According to your suggestion, we have added brief descriptions of these terms to make them easier to understand (Line 85-88).

As for the correlation between low motility and rejuvenation, we admit that it is speculative, as you commented. However, we think that the low motility of *T. rubra* may make it difficult for them to escape from unfavourable conditions quickly when the environment is subjected to drastic and adverse changes (e.g. food deprivation). Thus, rejuvenation into the diapause state as a cyst to tolerate unfavourable environment and remerge at a suitable time may be one of the survival strategies to preserve their populations in the face of environmental degradation.

Some interesting research on motility and survival strategies argues that adaptive motility changes in response to environmental conditions plays a key role in the survival of species, especially when food is depleted (Cho and Kim, 2013). The trade-off between reproduction and mobility prolongs organisms' survival and can be effective in protecting individuals from death due to lack of energy (Menezes and Rangel, 2023). There are two kinds of responses of biological organisms when almost all available food is consumed. The first one is a change in metabolism so that the organism can reduce the use of energy and wait for a better environment. The second response is increasing motility in order to find a place with food. This is quite similar to the cyst stage of *T. rubra*, and we therefore suggest that the low motility of *T. rubra* may lead them to choose the first response mode when they encounter unfavourable environmental mega-variations, lowering their metabolism, rejuvenating as cysts, and waiting for the favourable conditions to grow again as a new polyp. In order to improve the rigour and logic of the manuscript, we have modified the original description in the revised version, adding more information to clarify our point (Line 88-92).

References:

Cho E, Kim YJ. Starvation Driven Diffusion as a Survival Strategy of Biological Organisms. *Bull Math Biol* **75**, 845-870 (2013).

Menezes J, Rangel E. Trade-off between reproduction and mobility prolongs organisms' survival in rock-paper-scissors models. *EPL* **142**, 47002 (2023).

results

Comment 9: L.117-120 phylogenetic reconstruction: please give some summary numbers of the final alignment used for this analysis - no. of genes, no of positions, missing data. I can't see this information here, or in the methods.

Response: Thank you for your comment, and we apologise for the omission of this information. We used Gblocks to de-gap each set of homologous genes, choosing a parameter of $-b5=h$ to allow for the inclusion of Gap sites that are half of the total sequence data, a total of 1990 genes, with a tandem length of 383,088 amino acids. These details were added to the revised manuscript (Line 561-562).

Comment 10: L.124-127: "may coincide with the emergence of the statocyst and the ability of jellyfish to swim freely" - apropos of a *Clytia/Turritopsis* clade being compared to a scyphozoan clade, what is the relevance of this remark?

Response: Thank you for your comment reminding us that we should not correlate differentiation times with traits. We have removed this sentence from the revised manuscript to improve the rigour of the study.

Comment 11: L.136 Figure 2a - it is not clear how the zoomed regions relate to the tentacle bulb/statocyst. Can they be boxed? The paragraph talks about "a small, stone-like object" but it is not clear if this is pictured or not (I don't think it is).

Response: Thank you for your comments. In response to your suggestion, we have boxed the tentacle bulb/statocyst area to make it easier for the reader to visualise. In addition, Figure 2a is intended to show the difference in the ciliary structure of hair cells on the two sensory organs, and the phrase "Statolith, a small, stone-like object" is only intended to introduce the relationship between the statolith and the hair cell, in order to better explain Figure 2b.

genome annotation methods

Comment 12: L.489-492: the choice of cnidarian species for homology-based analysis appears inconsistent e.g. *Turritopsis* includes an anthozoan (*P.damicornis*) and a box jellyfish but *Aurelia* does not. Why ?

Response: When selecting cnidarian species for homology-based analyses, we chose species that are as closely related to the annotated species as possible. When we annotated the genome in 2021, there were many published genomes of scyphozoans, but there were basically no available genomic data in the class Hydrozoa, except for *Hydra* and *Clytia hemisphaerica*. Therefore we chose the representative species from each of Anthozoan, Hydrozoan, Cubozoan, and Scyphozoan to annotate the genome of *Turritopsis rubra*. In addition to the homology-based approach, *de novo* gene prediction programs and RNA-seq data from different tissues and life stages of the *T. rubra* were also used to predict the gene model, and the combination of these approaches makes our annotation results more convincing.

Comment 13: L.498: why was MAKER used for *A.coerulea* but not *T.rubra* ?

Response: Thank you for your comment. Because of the different availability of samples, the sequencing and annotation of these two species were performed at different times and by different companies, resulting in differences in the software used. However, we re-evaluated the annotation results in our subsequent analyses and tried to avoid interference from different sequencing and annotation methods in our comparative genomic analyses to minimise the impact of differences in sequencing and annotation methods.

Comment 14: Supplementary methods, genome annotation L.64: embryophyta_odb10 for a BUSCO database is clearly inappropriate as this is a plant dataset.

Response: Thank you for pointing out the inappropriateness of the database we used for the evaluation of the genome annotation results. We apologise for the lack of rigour in this section due to the direct use of data reported by sequencing companies. We have reevaluated the annotation results using the metazoa_odb10 database and corrected the result in the revised Supplementary Information (Supplementary Table 12).

Reviewer #2 (Remarks to the Author):

Summary:

Dong et al. have sequenced and compared two high-quality chromosome-level genomes of jellyfish with other cnidarian genomes, focusing on the genes related to the statocysts and hair cells of species with different swimming patterns during the evolution of jellyfish locomotion. They further employed transcriptomic and scRNA-seq analyses and found that genes involved in motile cilia and hair cell function were lower expressed in the sensory organs and hair cells of *Turritopsis rubra*. These results revealed the genetic basis for the absence of jellyfish balance organs, and the discovery of a potential gene family “OM” is quite interesting from a developmental and genomics evolutionary perspective. Generally, the manuscript is well-crafted, demonstrating considerable efforts to investigate the genetic foundation of jellyfish locomotor characteristics and survival strategies. The data presented in this study would enhance our understanding of the evolutionary lineages of movement and adaptation within the animal kingdom. I have a few clarifying questions and some comments aimed at enhancing the manuscript.

Response: Thank you for your complimentary remarks. We appreciate your insightful and helpful comments, which have greatly helped to improve our manuscript in this revision. We have addressed all your specific comments and provided a point-by-point response below.

Primary comments:

Comment 1: This study mainly focuses on the genetic basis of the absence of statocyst in *Turritopsis rubra*, however the title refers to the “specialised swimming pattern”, is there a direct/or indirect link between statocyst and jellyfish swimming pattern? The authors should give more explanation and provide some references.

Response: Thank you for your insightful comment. A reduction in the number of rhopalia, either through damage or experimental excision, also reduces the overall rate and regularity of swimming (Satterlie 2018). Specifically, the statocysts of the rhopalia in free-swimming jellyfish works as a feedback system that senses gravity and regulates the orientation of the body to control swimming. Their removal results in the loss of orientation and the inability of the jellyfish to perform righting movements (Mackie 1980). Thus, the possible link between the absence of statocysts and the special straight swimming pattern of *Turritopsis rubra* provides us with an interesting perspective for studying animal locomotion. We have added a few sentences to the introduction of the revised manuscript to explain and transition the relationship between statocysts and jellyfish swimming patterns (Line 73-75; 77-78).

References:

Mackie GO. Slow Swimming and Cyclical “Fishing” Behavior in *Aglantha Digitale* (Hydromedusae: Trachylina). *Can J Fish Aquat Sci* **37**, 1550-1556 (1980).
Satterlie R. Jellyfish locomotion. Oxford Research Encyclopedia of Neuroscience,

(2018).

Comment 2: This study identified a number of nerve and muscle related genes in the comparative genomes and also analyzed and discussed in the single cell comparisons of nerve and muscle cells, why did you study them? A brief explanation of their selection will shed light on their relevance to the research objectives.

Response: The regulation of jellyfish swimming mode is a very complex process that requires the coordination of muscles, the nervous system, and sensory organs. Nerve and muscle cells are pivotal in mediating the swimming behaviour of cnidarians. Nerve cells are crucial for the coordination and transmission of signals that control movement, and muscle cells are directly involved in the physical aspects of swimming. By analysing genes related to nerve and muscle cells, we aimed to uncover the genetic and cellular mechanisms that govern the distinct swimming patterns observed in *Turritopsis rubra*. This focus allowed us to explore specific genetic and cellular attributes contributing to the unique locomotive capabilities of this species, thus directly aligning with our research goals. We have added a brief explanation to the manuscript (line 268-271).

Comment 3: Single-cell transcriptomic analyses revealed the enrichment of diapause formation-related genes in the cyst during the reverse development of *Turritopsis rubra*, were any genomic comparisons made to support this at the genomic level? In addition, you mentioned to the fact that there have been comparative genomic analyses to study the reverse development of two *Turritopsis*. Have you also made genome comparisons in your study to obtain similar or opposite conclusions? This issue should also be discussed.

Response: Thank you for your insightful query regarding the scope of our research on *T. rubra*. Our single-cell transcriptomic analyses indeed indicated the enrichment of diapause formation-related genes in the cyst during the reverse development of this species. However, in this particular study, we did not extend our analyses to include genomic comparisons. The primary focus was on exploring the transcriptomic landscape, which, while revealing transcriptome data at the gene expression level, does not encompass genomic comparisons that could provide additional insights into genomic variations or structural changes.

Moreover, while we acknowledge that comparative genomic analyses have been instrumental in studying the reverse development of *Turritopsis* species, such genomic comparisons were not a part of this study. Our research was designed to delve into the transcriptomic aspects, thereby providing insights into the gene expression patterns and cellular processes during reverse development.

The points you raised highlight important avenues for future research. Genomic comparisons could significantly augment our understanding of the genetic basis of diapause and reverse development in *T. rubra*. Such analyses would be invaluable in determining whether the transcriptomic changes observed are underpinned by genomic alterations or are solely a result of differential gene expression.

In summary, while our current study provides a comprehensive view of the

transcriptomic changes during reverse development in *Turritopsis rubra*, integrating genomic data in future investigations would be a logical and valuable extension of this work, offering a more complete picture of the genetic and molecular mechanisms at play.

Comment 4: Some methods need improvements or clarification. For example, positive selection analyses were identified solely using the branch-site model in PAML, however, multinucleotide mutations can cause false inferences of lineage-specific positive selection. So, BS + MNM test were suggested to detect positive selection. Also, it is unclear how reliable of the lost genes found in this study.

Response: Thank you for your comment. We have conducted the BS + MNM test to detect the PSGs as you suggested, the result are provided in the Supplementary Data 5 and the text and figures of the manuscript have been updated based on the newly generated data. The key genes presented in our revised manuscript are now subjected to the PAML multiple test as well as the BS+MNM test, which gives them a great deal of credibility.

Multiple approaches have also been used to validate the reliability of the gene loss analysis. We have performed homology analysis in *de novo* assembled transcriptomes and genomes of related species to support our claims. The quality of our genomes was verified to rule out the potential sequencing bias, and confirm assembly integrity, gene annotation integrity, etc. We also conducted a PCR experiment on the key genes to verify that they were indeed present in the jellyfish containing the statocysts, whereas the corresponding bands could not be amplified in *T. rubra*. Details on this section are provided in the response to Reviewer 3 (Comment 1).

Secondary comments:

Comment 5: References are missing for some softwares, e.g., The amino acid substitutions of PSGs in different species were compared using MEGA-X v10.1.8. CellRanger v1.3 software pipeline, following the manufacturer's guidelines.

Response: Thanks for pointing out the omission of references for some software. We have added reference information to the revised manuscript (line 632 and Line 743).

Comment 6: Line 66-68: add specific information on the calibration points.

Response: We set a 240 Mya lower constraint on the *Acropora–Nematostella* split, as the first scleractinian corals appeared in fossil record after 240 Mya, and fossil calibrations were used with reference to the reported literature. The details have been added to the “Phylogenetic analysis” section of the Methods in the revised manuscript (Line 569-571).

Comment 7: Line 186: I think “CEP141” should be “CFAP141” according to the pictures and discussion?

Response: We changed “CEP141” to “CFAP141” in the revised manuscript (Line 199, 216, 239).

Comment 8: Line 203: again, CFAP141.

Response: We changed “CFAP14” to “CFAP141” in the revised manuscript (Line 199, 216, 239).

Comment 9: Line 252: the phrase "substantial enrichment" is somewhat vague.

Response: We have revised the manuscript to replace the term with “considerable enrichment” (Line 282).

Comment 10: Line 270: the use of "changed" is preferred over "altered" for a clearer implication of variance in the study's context.

Response: We have replaced the “altered” with “changed” (Line 300).

Comment 11: Line 286-288: add the relevant literature to support your claims.

Response: We have added relative references to support this claims. Related references were also added to the Reference section (line 313).

Comment 12: Line 371-372: given the complexity of Wnt, mTOR, Hippo, and FoxO signaling pathways, please provide specific examples where these pathways have been implicated in tissue regeneration or dormancy germination across different species.

Response: Thank you for your constructive feedback. In response to your suggestion, we have incorporated specific examples that elucidate the roles of the mentioned signalling pathways in the context of tissue regeneration and dormancy germination across a variety of species. These additions can be found in the revised manuscript in lines [402-408].

Comment 13: Line 420: polyps.

Response: Thank you for pointing out the spelling error, we corrected it in the revised manuscript (Line 462).

Comment 14: Line 420: “Ephyra” should be “ephyrae”.

Response: Thank you for pointing out the spelling error, we corrected it in the revised manuscript (Line 463).

Comment 15: Line 490: the correct form of *Aurelia* sp1 should be *Aurelia* sp.1.

Response: Thank you for pointing out this omission, we have corrected this in the revised version of the manuscript (Line 533).

Comment 16: Line 507: *T. rubra*

Response: Thank you for pointing out the spelling error, we modified it in the revised manuscript (Line 550).

Comment 17: Line 543: provide version information for DeepTE.

Response: Thanks for pointing out the omission, and we have added this information

to the revised manuscript (Line 590).

Comment 18: Line 624: 1E-5.

Response: Thank you for pointing out the spelling error, we corrected it in the revised manuscript (Line 629).

Comment 19: Line 646: are the cells retained on or passed through a 40- μm strainer?

Response: The cells were passed through a 40 μm strainer. To make this process clearer in our manuscript, we have modified the sentence to explicitly state the procedure. The revised sentence now reads: “After dissociation, the single-cell suspension was centrifuged at 500 x g for 5 min at 4 °C, resuspended in pre-chilled CMFASW, and then passed through a 40 μm cell strainer (FALCON, Corning, Corning, NY, USA).” (line 732-738).

Comment 20: Line 665: ensure consistency in the reference to functions. It is helpful to have them in quotes, and might also help to add the word “functions” after “FindNeighbors” and “FindClusters”.

Response: We agree that maintaining a uniform style in such references aids in clarity and readability. We have made the following changes: The term "FindNeighbors" has been revised to “the “FindNeighbors” function”, and similarly, "FindClusters" has been updated to “the “FindClusters” function”. These changes ensure a consistent and clear reference to the specific functions used in our analysis. These revisions can be found in the manuscript at line [755-757].

Comment 21: Line 647: How to determine it?

Response: Calcein AM and Draq7TM are fluorescent dyes commonly used in cell viability assays. Calcein AM is a non-fluorescent, cell-permeant compound that is converted to a green fluorescent calcein when hydrolysed by intracellular esterases in live cells. This fluorescence is a marker of live cells, as these esterases are active in viable cells. On the other hand, Draq7TM is a cell-impermeant dye that only enters cells with compromised membrane integrity, typical of dead or dying cells, and fluoresces red upon binding to DNA.

In our study, low concentrations of Calcein AM (2 mM) and Draq7TM (0.3 mM) were used to stain the cells. The presence of green fluorescence indicated live cells, while red fluorescence indicated non-viable cells. This dual-staining method allowed us to accurately assess the proportion of live and dead cells in our samples. We have added a brief description of the detection method to the revised manuscript (line 735-738).

Comment 22: Line 655-657: why you setting this criteria? Is there any citations?

Response: During cell capture, some cells may be compromised, leading to RNA degradation and low-quality sequencing data characterised by reduced gene and UMI counts. These are classified as low-quality cells. In parallel, technical artifacts such as doublets or multiplets—where two or more cells are inadvertently captured together—

can result in misleadingly high gene and UMI counts. To circumvent these issues, we have established filtering criteria based on the gene and UMI count distributions in the raw data. By applying these filters, we have enhanced the quality and reliability of the data, which is crucial for the downstream bioinformatic analysis and biological interpretations. we have updated the Reference section with pertinent literature that supports our approach (lines 748-749).

Comment 23: Line 710: Verify the accession code PRJNA1014582, as the datasets are currently not accessible.

Response: Thank you for bringing up this issue regarding the accessibility of the supplementary datasets and the BioProject citation. All sequencing data has been uploaded to NCBI and the BioProject number has been processed; we will release the data as soon as the article is confirmed to be accepted (Line 800).

Comment 23: The indicator lines in Figure 1b are not in the correct position, check and correct them.

Response: Thank you for pointing out the error in the image, we have corrected it in the revised manuscript.

Reviewer #3 (Remarks to the Author):

Summary:

In this manuscript, Dong and colleagues sequenced one new hydrozoan and one new scyphozoan genomes, analysed the expansion and loss of genes, and carried out single-cell transcriptomic analyses of five developmental stages. Despite the field of animal genomics has rapidly advanced in last two years and obtaining new genomic resources have different weight nowadays, I think this manuscript has intriguing results in its gene loss and single-cell data analyses.

Response: We appreciate your insightful and helpful comments, which have greatly helped to improve our manuscript in this revision. We have addressed all your specific comments and provided our responses below.

Primary comments:

Comment 1: Regarding the gene loss analyses, it is generally well known to be difficult to prove in the field, and the authors should provide multiple lines of evidence to support their claims. For instances, are these genes also not contained in the transcriptomes in respective and related species? How about genomes of other related species that are not included in the current analyses, can the authors look into them and ensure they are also not there? Other additional analyses could also be included such as syntenic and gene pathway (than individual gene) analyses. These will make the claim much more solid.

Response: We agree that analyses on gene loss are hard to prove, and based on your comments, we have performed more analyses on the transcriptome and related genomes to make sure that they are also absent in related species. A PCR experiment was also conducted on the key genes to verify that they were indeed present in the jellyfish possessing statocysts, whereas the corresponding bands could not be amplified in *T. rubra*.

1. **Target sequences alignment to the *de novo* assembled transcriptome:** The raw transcriptome sequencing data for the related species *Turritopsis dohrnii* was downloaded from NCBI (72 samples), and the raw sequencing data for *Turritopsis rubra* transcriptomes (8 samples) in our study were used for the analyses. Trinity (v2.13.2) software was used to perform de novo assembly of each sample separately to obtain the transcripts, and tblastn was used to compare the target gene sequences with the transcripts of *T. dohrnii* and *T. rubra* to exclude the possibility of gene loss due to assembly.

Target sequence alignment to our assembled genome of *T. rubra* and three other assemblies of the genus *Turritopsis*: The sequence data for three published *Turritopsis* genomes (Gold et al, 2019; Khalturin et al, 2019) were downloaded from the NCBI: *Turritopsis dohrnii*_S001 (GCA_025167195.1), *Turritopsis dohrnii*_TUR (GCA_027922465.1) and *Turritopsis rubra* (GCA_025167575.1). The target gene sequences were compared to genomic DNA sequences of four assemblies using tblastn

to rule out the possibility of gene loss due to gene prediction.

The results of sequence matching and the distribution of blast score values indicate that these loss genes are indeed absent from *T. rubra* and related species from *Turritopsis*. Note that some similar sequences were compared in CFAP141, but the score values were lower than the reference sequences and were not considered to have screened for homologous genes. Sequence matching results files and distribution maps of blast score values are uploaded separately as support for the responses (GeneLoss_Blast.xlsx).

Read alignment to our genome was corrected to verify assembly quality: The results of the comparison between raw data from PacBio sequencing and the homologous genes of the six loss genes are shown in the file Subreads.blast.result.xls. The low score indicated that no candidate sequences were screened in the raw data. The analysis of the raw sequencing data shows that our genome assembly is continuous, and it is unlikely that genes will be lost due to assembly problems.

2. PCR experiment: In order to verify the presence of the genes of interest in the related species, we performed PCR on six species, including *T. rubra* and five species with statocysts, using the corresponding primers for the candidate genes. Three adult

individuals of each species were selected to extract the DNA and obtain the PCR templates. We tested the success of these reactions by performing electrophoresis of the resulting products on a 1.5% agarose gel. Universal primers for cnidarian 16S were used as a control to demonstrate the validity of DNA templates. Because the species are distantly related and the sequences of the target genes are not well conserved, it was not possible to design universal primers, so specific primers were designed individually for each species, and primers for all species were added to the PCR experiments for *T. rubra*. The results showed that the listed loss genes were present in jellyfish possessing statocysts, but no corresponding sequence bands were amplified in *T. rubra*.

Reviewer #4 (Remarks to the Author):

Summary:

Dong et al. conducted a study investigating the comparative genomic and single-cell transcriptomic analyses of two jellyfish species, namely *Turritopsis rubra* and *Aurelia coerulea*. They performed *de novo* assemblies of two reference genome and identified several candidate genes associated with swimming patterns, specifically PSGs. Additionally, they performed comparative cellular analyses across two species and five developmental stages of *Turritopsis rubra*, leading to the identification of multiple candidate different expressed genes in hair cells (e.g., PSGs) that may be involved in swimming patterns, and the diapause formation-related genes in the cyst during the reverse development of *Turritopsis rubra*, respectively. While this manuscript presents an important resource for studying the movement patterns and shape of jellyfish, there are a few areas that require further clarification and more robust analysis.

Response: We appreciate your insightful and helpful feedback, which greatly helped to improve our manuscript in this revision. And we have addressed all your specific comments and provided our point-by-point responses below.

Major comments:

Comment 1: In the section of sequencing and genome assembly, it is mentioned that *A. coerulea* was directly sequenced using Pacbio Hifi CCS technology. However, it was not clearly stated which sequencing technology was used for *Turritopsis rubra*. Was Pacbio CLR or Hifi CCS employed for this species? It is also important to provide information about the assembly software or pipelines used, including version numbers and parameters. Additionally, an explanation is needed for why the Hi-C data of the two species were processed using different pipelines. It is crucial to evaluate any potential issues that may arise from using different sequencing technologies or assembly pipelines. For instance, it is worth investigating why the contig N50 of *Aurelia coerulea* assembly is notably longer than that of *Turritopsis rubra*, and why the complete BUSCOs of *Aurelia coerulea* exhibit relatively lower completeness and higher fragmentation compared to *Turritopsis rubra*.

Response: Thank you for your comments pointing out the confusing aspects of our sequencing methodology and bioconfidence analysis. PacBio CLR was employed for *T. rubra*, and the details of the assembly software or pipelines are provided in the revised manuscript (Lines 506-507).

Because of the different availability of samples, the sequencing and assembly of these two species were performed at different times and by different companies, resulting in differences in the pipelines used. However, we evaluated these assemblies to ensure that they were of high quality. In subsequent analyses, we also tried to avoid the interference of different sequencing methods in comparative genomic analyses as much as possible to minimise the impact caused by methodological differences. From Contig N50, the sequence continuity of *A. coerulea* genome is a bit higher than that of

T. rubra, but the genome of *A. coerulea* is longer by 566 Mb; that of *T. rubra* is only 266 Mb. The metazoa_odb10 database that we used for evaluation has reference sequences of *Hydra vulgaris*, which is more closely related to *T. rubra* evolution and may be more likely to possess homologous sequences, and thus the complete BUSCOs for *T. rubra* may be higher than those for *A. coerulea*.

Comment 2: The author employed different single-cell sequencing platforms, namely 10X Genomics and BD Rhapsody, to generate single-cell transcriptomic data for *T. rubra* and *A. coerulea*. However, the potential batch effect resulting from using these different technologies in the comparative cellular analysis should be thoroughly evaluated. Additionally, the author mentioned the dynamic changes in cell composition proportions across different developmental stages of *T. rubra*, which necessitates increased biological replication due to potential bias introduced by two scRNA-seq technologies and individual differences. These factors can significantly impact cell composition and consequently influence the conclusions drawn from this study. Furthermore, accurate classification of cell types is a critical and challenging task for two non-model species. Therefore, the author should provide details regarding their approaches to defining cell types and assess the annotation accuracy of cell types in both species across the five different developmental periods of *T. rubra*.

Response: We appreciate your feedback as it has significantly contributed to the enhancement of our methodological rigor and the overall quality of our study.

1. Addressing batch effects: You're absolutely right about the importance of thoroughly evaluating the batch effects that might arise from employing different technologies like 10X Genomics and BD Rhapsody in our comparative cellular analysis. To address this, we have implemented the “Harmony” package for batch correction and normalisation to mitigate these effects. Canonical correlation analysis (CCA) is a popular algorithm for integrating single-cell RNA sequencing (scRNA-seq) data from different conditions or platforms to reduce or eliminate batch effects. It improves the quality of data integration by adjusting cell characteristics in the data so that similar cell populations from different batches or conditions are closer together in multidimensional space. We have added the reference to the Reference section (line 780).

2. Biological replication for *T. rubra*: Regarding the concern about biological replication, we understand the importance of minimising bias in our study. However, given the unique challenges posed by the rarity and the stringent collection conditions necessary for these non-model species, coupled with the complexities of inverse development and the hurdles for obtaining samples at synchronous developmental stages, it was not feasible to procure more biological samples. We have emphasised this limitation in our discussion (line 417-423), highlighting the need for cautious interpretation of the data. Despite this limitation, we believe the findings provide valuable preliminary insights and set a foundation for future studies when more samples become available.

3. Classification and annotation of cell types: The accurate classification and annotation of cell types in non-model species such as *T. rubra* and *A. coerulea* present

significant challenges. To address this, we first aligned samples from *A. coerulea* and *T. rubra* by comparing the expression similarity of each cluster using CCA, a method implemented in Seurat. To mitigate batch effects across samples and experiments, particularly those involving different developmental stages in *T. rubra*, we employed the “Harmony” package. This approach ensured consistency in our data analysis, minimizing technical variations while preserving biological significance.

Following alignment and batch effect correction, we utilised a suite of Seurat functions for cell clustering, grouping, and the identification of differentially expressed genes. Our initial step in cell type determination involved the use of known cell type-specific or enriched marker genes from *Aurelia*, as previously described in the literature (Gold et al., 2019; Li et al., 2024). Building on this foundation, we expanded our annotation strategy to include marker genes identified in closely related species such as *hydra*, *Clytia hemisphaerica*, and *Nematostella vectensis* (Chari et al., 2021; Siebert et al., 2019; Sebe-Pedros et al., 2018; Li et al., 2024). By integrating these markers with cluster analysis based on gene expression patterns, we can more accurately annotate cell types, enhancing the depth and reliability of our findings. We have detailed our methodologies and analytical strategies in the Methods section of our manuscript (line 752-761), with all pertinent references cited in the Reference section (line 1027-1034).

References:

- Tim S, *et al.* Comprehensive integration of single cell data. *Cell* **177**: 1888-1902.e21 (2019).
- Chari T, *et al.* Whole-animal multiplexed single-cell RNA-seq reveals transcriptional shifts across *Clytia* medusa cell types. *Science Advances* **7**, 7(48): eabh1683 (2021).
- Siebert S, *et al.* Stem cell differentiation trajectories in Hydra resolved at single-cell resolution. *Science* **365**: eaav9314. (2019).
- Gold DA, Lau CLF, Fuong H, Kao G, Hartenstein V, Jacobs DK. Mechanisms of cnidocyte development in the moon jellyfish *Aurelia*. *Evol Dev* **21**: 72-81 (2019).
- Sebe-Pedros A, *et al.* Cnidarian Cell Type Diversity and Regulation Revealed by Whole-Organism Single-Cell RNA-Seq. *Cell* **173**: 1520-1534 (2018)
- Li Y., *et al.* Molecular and cellular basis of life cycle transition provides new insights into ecological adaptation in jellyfish. *The Innovation Geoscience* (2024) (submitted)

Minor comments:

Comment 1: Figure 2a is positioned before Figure 1d in the text and requires adjustment in terms of layout. And the two ordinate labels in Figure 1d are placed closely together, potentially causing confusion. It is recommended to adjust and differentiate them.

Response: Since the scanning electron microscope image in Figure 2a corresponds to the hair cell schematic in Figure 2b, we have adjusted the order of some of the paragraphs according to your suggestions for the overall presentation of the images and the logic of the manuscript (Line 153-159). As for Figure 1d, we have widened the distance between the two ordinate labels and added a separator line in the middle to

better distinguish between them.

Comment 2: The scale in Figure 2a appears too small and blurry, making it difficult to view clearly.

Response: We appreciate your suggestion to improve the clarity and comprehensibility of this figure. In order to balance the readability of Figure 2a with the appearance of the overall layout of Figure 2, we have appropriately enlarged Figure 2a and adjusted the font size for a better presentation of the information.

Comment 3: In lines 140-143, the authors identified a total of 548 PSGs. Considering the relatively large divergence time of these species, the authors should carefully examine the alignment results of selective amino acid loci and their upstream and downstream sequences in these PSGs. This is important to mitigate the risk of higher false positives in PSG identification.

Response: We have performed multiple testing adjustments on the PSGs. The alignment results of selective amino acid loci in PSGs and their upstream and downstream sequences were carefully examined and rigorously screened during the analyses to reduce the risk of false positives in PSG identification.

Comment 4: In lines 186-187, the authors mentioned in the manuscript that "the loss of CFAP141 and CEP97 in *T. rubra* may also result in the absence of Statoliths," while "CFAP141" is marked in the figure. This appears to be a spelling error that needs correction.

Response: Thank you. We have changed "CEP141" to "CFAP141" in the revised manuscript (Line 199, 216, 239)

Comment 5: It is important to assess the loss genes shown in Figure 3a, including CFAP141, CEP97, KCNK1, NSMF, and MAOA, due to the fact that they exhibit only 1-2 gene copies in other comparative species. The evaluation should consider potential sequencing bias, assembly integrity, gene annotation integrity, or use PCR experimental data.

Response: Thank you for your comment alerting us to potential issues regarding sequencing bias, assembly integrity, and gene annotation integrity that may limit confidence in gene loss results. We have therefore added some analyses involving the transcriptome, related genomes, and a PCR experiment to address these issues in response to your suggestions, and the details are provided in the response to Reviewer 3 (Comment 1).

Comment 6: In lines 193-195 and Figure 3b, considering the substantial divergence time between the used species (from jellyfish to mammals), it is necessary to examine the alignment results of upstream and downstream amino acid sequences of these conserved sites.

Response: Thank you for your comments. We have carefully examined the alignment results of upstream and downstream amino acid sequences of these conserved sites, and

the results are provided below:

CHSY1

```

TurritopsisRubra  K Y H R H I G R N R R R M T V
ClytiaHemisphaerica  K Y H R H V G Y N R R R M T V
MorbakkaVirulenta  K Y H R H I G Y N R R R M T V
SanderiaMalayensis  K Y H R H I G Y N R R R M T V
AureliaCoerulea  K Y H R H I G Y N R R R M T V
CassiopeaXamachana  K Y H R H I G Y N R R R M T V
NemopilemaNomurai  K Y H R H I G Y N R R R M T V
RhopilemaEsculentum  K Y H R H I G Y N R R R M T V
  DanioRerio  L Y K K H K G --- K T M T V
  GallusGallus  L Y K K H K G --- K K M T V
  MusMusculus  L Y K K H K G --- K K M T V
  HomoSapiens  L Y K K H K G --- K K M T V
                    475           480           485

```

USH2A

```

TurritopsisRubra  C D N K G T F Q N S G E C N Q
ClytiaHemisphaerica  C D S K G T V G N T G Q C D Q
MorbakkaVirulenta  C D L K G I V G G S A N C N Q
SanderiaMalayensis  C D S K G T Q G N S G L C N Q
AureliaCoerulea  C D S K G T V G H A G Q C N A
CassiopeaXamachana  C D V Q G T V G N T G Q C D F
NemopilemaNomurai  C D S K G T Q G N S G Q C N P
RhopilemaEsculentum  C D T K G T Q G N T G Q C N P
  DanioRerio  C D L R G T V N G S G V C D K
  GallusGallus  C D K A G T V N G S L L C D K
  MusMusculus  C E K M G T V N G S L R C D K
  HomoSapiens  C D K T G T I N G S L L C N K
                    935           940           945

```

CDH23

```

TurritopsisRubra  H F E I V V N I K D G G H P P
ClytiaHemisphaerica  S H S I I V L A T D K G T P S
MorbakkaVirulenta  E F S A V I T A S D T G K P S
SanderiaMalayensis  H L S V A I F A S D Q G V P A
AureliaCoerulea  F Y K L I L I A L D Q G K P P
CassiopeaXamachana  R Y I L V L V A V D Q G K P P
NemopilemaNomurai  V Y Q L E V I A A D Q G K P P
RhopilemaEsculentum  V Y Q L E V I A V D Q G K P P
  DanioRerio  Y Y N I T I T A K D L G T P S
  GallusGallus  F Y N L T I S A R D R G V P P
  MusMusculus  F Y N L T I C A R D R G V P P
  HomoSapiens  F Y N L T I C A R D R G M P P
                    2012           2016           2020           2024

```

DCTN1

```

TurritopsisRubra  S D A R R D C R E F L A D S L
ClytiaHemisphaerica  A S A E R D C T D F L S D S L
MorbakkaVirulenta  A S H P V S C T D Y L V E A M
SanderiaMalayensis  G A H P L S E T E Y L W D A I
AureliaCoerulea  G S H Q L S K T E F L N D A I
CassiopeaXamachana  G S Y P V S Q T E F L N D A I
NemopilemaNomurai  G N Y E N S Q T E F L N D A I
RhopilemaEsculentum  G N F E I S Q T E F L N D A I
  DanioRerio  A E Q P E D C T M Q L A D H I
  GallusGallus  A E Q A E D C T M Q L A D H I
  MusMusculus  A E Q P E D S T M Q L A D H I
  HomoSapiens  A E Q P E D C T M Q L A D H I
                    1086           1090           1094

```

CEP83

```

TurritopsisRubra  D S H K L R M A Q R E N T Q L
ClytiaHemisphaerica  D V Q R L R S L Q K E N A Q L
MorbakkaVirulenta  D S Q K M R I L Q R D N A Q L
SanderiaMalayensis  D T Q R I R V L Q K D N A Q L
AureliaCoerulea  D A Q R I R V L Q K D N A Q L
CassiopeaXamachana  D A Q K I R I L Q K D N A Q L
NemopilemaNomurai  D A Q R L R T L Q K E N A Q L
RhopilemaEsculentum  D A Q R L R T L Q K E N A Q L
  DanioRerio  D G K R V E A L L R E K A Q L
  GallusGallus  D S K R V E V L S R E K A Q L
  MusMusculus  D S K R M E Q L V R E K T H L
  HomoSapiens  D S K R V E Q L A R E K V Y L
                    225           230           235

```

KIAA2026

```

TurritopsisRubra  L N D E L V Y G Y R I F R E L
ClytiaHemisphaerica  L T D G L R Y G Y K V F R E L
MorbakkaVirulenta  L T P E L R H A Y R I C R E M
SanderiaMalayensis  M S P E I R H A Y R I F R D M
AureliaCoerulea  M S P E L R H A Y R I F K E M
CassiopeaXamachana  M T P E V R S A Y K I F R E L
NemopilemaNomurai  M S P E I R H A Y K I L R E I
RhopilemaEsculentum  M S P E I R H A Y K I L R E I
  DanioRerio  L K P E V Q Q A H R I F Q S F
  GallusGallus  L S P E L Q Q G Y R I L R E F
  MusMusculus  L T Y E L Q Q G Y R I L G E F
  HomoSapiens  L T Y E L Q Q G Y R I L G E F
                    132           136           140           144

```

Comment 7: The in situ hybridization results in Figure 4c appear to be fuzzy and unclear. It is recommended to replace this image with a high-definition picture to improve visibility.

Response: Thank you for pointing out the blurriness of the image; the pixels may have been compressed during the process of combining the images. We have replaced this with higher-resolution picture to improve visibility. Due to the thickness of the biological samples, when focusing on the colouring points, other parts may look blurred due to the fact that they are not in the same layer of focus, but this will not affect the information that we want to present in the pictures.

Comment 8: In Supplementary Figure 5, the overlapping and illegible names of the samples needs to be addressed for improved readability.

Response: Thank you for pointing out the oversight in the figure; we fixed the name overlap and replaced the label with a more readable one.

Comment 9: In Supplementary Figure 10a, the color of the cell type numbers does not correspond to the color of the cells themselves. This discrepancy should be rectified for

consistency.

Response: Thank you for bringing this discrepancy in Supplementary Figure 10a to our attention. We agree that consistency between the colour of cell type numbers and the corresponding cell colours is crucial for clear and accurate representation of the data. We have revised the figure to ensure that the colours of the cell type numbers now correctly correspond to the colours of the cells themselves. This amendment enhances the figure's clarity and ensures accurate visual representation of the cell types. The updated figure can be found in the revised Supplementary Information (Supplementary Figure 11a).

Comment 10: In Supplementary Figure 10b, certain marked genes are not unique to specific cells. For example, the gene MEIG1 is marked for both cells 8 and 2.

Response: Thank you for pointing out the issue in Supplementary Figure 10b regarding the non-unique marking of certain genes across different cell types. Upon re-examination of the figure, we acknowledge that the gene MEIG1 was indeed marked for both cell types 8 and 2, which could lead to misinterpretation. To address this, we have carefully reviewed our dataset and revised the figure to accurately reflect the unique gene markers for each cell type. The updated Supplementary Figure 11b now correctly displays distinct gene markers for each cell type, ensuring that each marker is exclusive to a single cell type. The updated figure can be found in the revised Supplementary Information (Supplementary Figure 11b).

Comment 11: It is suggested to use a semicolon to indicate numerical values in the supplementary tables, with two decimal places retained. For instance, Supplementary Tables 11 and 14 should follow this formatting recommendation.

Response: We have refined the format of these supplementary tables based on your suggestions to make them more standardised.

REVIEWER COMMENTS

Reviewer #2 (Remarks to the Author):

The author has addressed most of my concerns, but one final concern remains regarding the genetic basis for the swing pattern. It remains unclear how the absent and positive-selection genes are involved in determining the swimming pattern. Additionally, the author only investigated the functional impact of OM genes on statocyst development using the siRNA approach. To provide a more comprehensive understanding, it would be beneficial for the author to integrate all the findings (including absent genes, PSGs, and single-cell data) and present a cohesive picture demonstrating how these genetic differences contribute to the swing pattern.

I suggest that the section discussing the "Genetic profiles of hair, neural, and muscle cells in *T. rubra* and *A. coerulea*" be either reduced or removed. This adjustment would allow the author to place a greater focus on exploring the relationship between genetic changes and single-cell data in relation to the swing pattern. By prioritizing this aspect, a more coherent and concise presentation of the research findings can be achieved.

The text still contains grammar errors that need to be addressed. For example, "We further conducted in situ hybridisation analysis was conducted to confirm whether these genes were involved in statocyst formation" should be corrected to "We further conducted in situ hybridisation analysis to confirm whether these genes were involved in statocyst formation". It is suggested that the manuscript be polished by a native speaker before publication to ensure such errors are corrected.

Reviewer #3 (Remarks to the Author):

In this revised manuscript, the authors have carried out de novo assemblies of transcriptomes from *Turritopsis* species, as well as checking other three other *Turritopsis* genomes, in order to address the claim of the gene loss concern as suggested in the previous round of comments. While this has certainly improved the confidence of the claim, and I personally also appreciate the effort, it is still necessary to carry out the syntenic analyses (together with non-*Turritopsis* genomes) as suggested in the previous round of comments, in order to ensure the genes are really lost rather than as very fast evolving genes.

Reviewer #4 (Remarks to the Author):

In this revised manuscript, the authors have effectively addressed and resolved my questions, with the exception of a few points. Obtaining samples at different developmental stages presents a challenge in achieving a single cell experiment with biological replication. The results presented in the manuscript regarding cell types and gene expression are credible. However, the conclusions drawn about changes in cell proportion at different developmental stages (Line 192-294) are highly uncertain. It is strongly recommended that these results should be removed from the manuscript. Additionally, I suggest that the authors can conduct the pseudotime analysis with single-cell RNA-sequencing at different developmental stages to better understand the relationship between cell type differentiation.

Point-by-point response to reviewer's comments

Reviewer #2 (Remarks to the Author):

Comment 1: The author has addressed most of my concerns, but one final concern remains regarding the genetic basis for the swing pattern. It remains unclear how the absent and positive-selection genes are involved in determining the swimming pattern. Additionally, the author only investigated the functional impact of OM genes on statocyst development using the siRNA approach. To provide a more comprehensive understanding, it would be beneficial for the author to integrate all the findings (including absent genes, PSGs, and single-cell data) and present a cohesive picture demonstrating how these genetic differences contribute to the swing pattern.

Response: Thank you for your insightful comments which made us realize the need for a comprehensive picture to present our findings more fully. Therefore, we have integrated the main results of our genomic and transcriptomic analyses (Figure 1) to show the absent genes, PSGs, and down-regulated or specifically non-expressed genes in the hair cells of *T. rubra*, which may be involved in statolith formation and normal hair cell function, both of which are essential for statocyst formation. As the statocysts of free-swimming jellyfish work as a feedback system that sense gravity and regulates the orientation of the body to control swimming and removal of statocyst results in the jellyfish losing its sense of orientation and being unable to perform righting movements (Satterlie, 2018). The genetic differences between the two representative jellyfish, i.e., *T. rubra* and *A. coerulea*, may contribute to the presence or absence of statocysts and different swimming patterns. This image is also provided in the revised manuscript as Extended Data Figure 8.

Figure 1. Genetic basis for the swimming patterns of *T. rubra* and *A.coerulea*. Lost genes (grey), positively selected genes (orange), and downregulated- or non-expressed genes and pathways (blue) in the hair cell of *T. rubra*, which are related to statocyst formation and cilium function, may result in the loss of statocyst and straight swimming patterns in *T. rubra*.

References:

Satterlie R. Jellyfish locomotion. Oxford Research Encyclopedia of Neuroscience, (2018).

Comment 2: I suggest that the section discussing the "Genetic profiles of hair, neural, and muscle cells in *T. rubra* and *A. coerulea*" be either reduced or removed. This adjustment would allow the author to place a greater focus on exploring the relationship between genetic changes and single-cell data in relation to the swing pattern. By prioritizing this aspect, a more coherent and concise presentation of the research findings can be achieved.

Response: Thank you for your comment. We have appropriately streamlined some discussion in this section according to your suggestions and have removed some results that are not directly relevant to the conclusions. We believe that now the possible relationship between the data and the swimming pattern is presented more clearly and simply in the revised manuscript (Lines 236-258).

Comment 3: The text still contains grammar errors that need to be addressed. For example, "We further conducted in situ hybridisation analysis was conducted to confirm whether these genes were involved in statocyst formation" should be corrected to "We further conducted in situ hybridisation analysis to confirm whether these genes were involved in statocyst formation". It is suggested that the manuscript be polished by a native speaker before publication to ensure such errors are corrected.

Response: Thanks to your reminder, we have invited a native speaker to re-touch up the revised manuscript and carefully correct all grammatical and spelling errors in order to keep the language of the manuscript regular and to enhance its readability. Grammatical corrections are highlighted in green in the text.

Reviewer #3 (Remarks to the Author):

Comment: In this revised manuscript, the authors have carried out *de novo* assemblies of transcriptomes from *Turritopsis* species, as well as checking other three other *Turritopsis* genomes, in order to address the claim of the gene loss concern as suggested in the previous round of comments. While this has certainly improved the confidence of the claim, and I personally also appreciate the effort, it is still necessary to carry out the syntenic analyses (together with non-*Turritopsis* genomes) as suggested in the previous round of comments, in order to ensure the genes are really lost rather than as very fast evolving genes.

Response: Thank you for your insightful comment, we agree with you that syntenic analyses are important to confirm gene loss and have performed the syntenic analyses with other jellyfish species according to your suggestion at the level of the genome and at the level of key genes.

1. Genomic syntenic analyses: The genomic synteny is not significant due to the small number of jellyfish species for which genomic data are currently available and the distant affinities of these jellyfish with *T. rubra*, which belong to separate orders (Figure 2). The positions of genes upstream and downstream of the key genes were also manually examined, and none of them were on the syntenic block.

Figure 2. The genomic synteny of *T. rubra* and other jellyfish.

2. Local gene syntenic analysis: The results showed that local gene synteny is more significant in jellyfish that are more closely related (scyphozoans), but not significant when compared to jellyfish that are distantly differentiated (Figure 3). Considering the distant divergence of species currently available for comparison, we suggest that using syntenic analyses to confirm gene loss may not be a very beneficial approach for *T. rubra*.

Figure 3. Local gene synteny of key candidate genes.

3. To further ensure that the genes are lost instead of being very fast-evolving genes or selected genes, we pulled out the amino acid sequences of the candidate genes in other jellyfish and performed amino acid comparisons with the whole genome sequence of *T. rubra* using BLASTp (E-value of $1e^{-5}$), and in the results, none of the corresponding sequences were found, which suggested that the candidate genes were indeed lost in the genome of *T. rubra*.

4. In addition, our gene loss analyses are based on the comparison of amino acid

sequences of selected species. Gene loss analyses in some teleost is generally set at a threshold of $1E^{-10}$ (Bian et al., 2016; Solbakken *et al.*, 2016; Policarpo *et al.*, 2021). However, in our analyses, considering the more distant divergent relationships of the jellyfish species, the threshold was set at $1E^{-5}$, which is broader compared to that used for teleost fish. However, at this broad threshold, homologous sequences of the candidate gene were identified in all other jellyfish but were still not found in *T. rubra*.

Combined with our analyses of the transcriptomes and genomes of *Turritopsis* species and the results of PCR validation of key genes in our previous response, and the functional validation result that the key lost gene OM indeed affects biological function, we believe that our results of gene loss in *T. rubra* are relatively convincing.

References:

- Solbakken M, *et al.* Evolutionary redesign of the Atlantic cod (*Gadus morhua* L.) Toll-like receptor repertoire by gene losses and expansions. *Sci Rep* **6**, 25211 (2016).
- Bian C, *et al.* The Asian arowana (*Scleropages formosus*) genome provides new insights into the evolution of an early lineage of teleosts. *Sci Rep* **6**, 24501 (2016).
- Policarpo M, *et al.* Evolutionary dynamics of the *OR* gene repertoire in teleost fishes: evidence of an association with changes in olfactory epithelium shape. *Mol Biol Evol* **38**, 3742-3753 (2021).

Reviewer #4 (Remarks to the Author):

Comment: In this revised manuscript, the authors have effectively addressed and resolved my questions, with the exception of a few points. Obtaining samples at different developmental stages presents a challenge in achieving a single cell experiment with biological replication. The results presented in the manuscript regarding cell types and gene expression are credible. However, the conclusions drawn about changes in cell proportion at different developmental stages (Line 292-294) are highly uncertain. It is strongly recommended that these results should be removed from the manuscript. Additionally, I suggest that the authors can conduct the pseudotime analysis with single-cell RNA-sequencing at different developmental stages to better understand the relationship between cell type differentiation.

Response: Thank you for your valuable feedback. In response to your suggestions, we have made the following modifications to our manuscript:

1. We have omitted the sections pertaining to changes in cell composition across different developmental stages to ensure the rigor of our conclusions.

2. To address the challenge of obtaining samples at various developmental stages for single-cell experiments with biological replication, we employed the LIGER (Liu et al., 2020) package for data integration across stages, aiming to minimise batch effects.

3. In response to the recommendation, we have expanded our analysis to include pseudo-time analysis of cell differentiation at different developmental stages using single-cell RNA-sequencing (Figure 4 and 5). This analysis was performed using a suite of pseudo-time analysis packages, including monocle 3, monocle 2, and cytoTRACE. The specifics of these analyses are detailed in the manuscript (Lines 286-302, Lines 385-394).

Figure 4. Cell differentiation trajectories at different stages in *T. rubra*.

Figure 5. Analysis of the cell differentiation trajectory at different stages in *T. rubra*.

4. The figures relevant to this analysis (Figure 5 and Extended Data Figure 7) have been updated. Additionally, we have enriched the manuscript by detailing the analytical methods and adding pertinent references in the Materials and Methods section (lines 791-798) and the References section, respectively.

Reference:

Liu J, Gao C, Sodicoff J, Kozareva V, Macosko EZ, Welch JD. Jointly defining cell types from multiple single-cell datasets using LIGER. *Nat Protoc* **15**, 3632-3662 (2020).

REVIEWERS' COMMENTS

Reviewer #2 (Remarks to the Author):

The author has successfully addressed all of my concerns, leaving me with no further questions.

Reviewer #3 (Remarks to the Author):

The authors have addressed my concerns on the claim of gene loss by performing additional analyses.

Reviewer #4 (Remarks to the Author):

The authors have addressed all of my concerns.